# HSV-1 single-cell analysis reveals the activation of anti-viral and developmental programs in distinct sub-populations

Nir Drayman[1,2]*, Parthiv Patel[1,2], Luke Vistain[1,2], Savaş Tay[1,2]*

[1]Institute for Molecular Engineering, The University of Chicago, Chicago, United States; [2]Institute for Genomics and Systems Biology, The University of Chicago, Chicago, United States

**Abstract** Viral infection is usually studied at the population level by averaging over millions of cells. However, infection at the single-cell level is highly heterogeneous, with most infected cells giving rise to no or few viral progeny while some cells produce thousands. Analysis of Herpes Simplex virus 1 (HSV-1) infection by population-averaged measurements has taught us a lot about the course of viral infection, but has also produced contradictory results, such as the concurrent activation and inhibition of type I interferon signaling during infection. Here, we combine live-cell imaging and single-cell RNA sequencing to characterize viral and host transcriptional heterogeneity during HSV-1 infection of primary human cells. We find extreme variability in the level of viral gene expression among individually infected cells and show that these cells cluster into transcriptionally distinct sub-populations. We find that anti-viral signaling is initiated in a rare group of abortively infected cells, while highly infected cells undergo cellular reprogramming to an embryonic-like transcriptional state. This reprogramming involves the recruitment of β-catenin to the host nucleus and viral replication compartments, and is required for late viral gene expression and progeny production. These findings uncover the transcriptional differences in cells with variable infection outcomes and shed new light on the manipulation of host pathways by HSV-1.
DOI: https://doi.org/10.7554/eLife.46339.001

*For correspondence:
nirdra@uchicago.edu (ND);
tays@uchicago.edu (SşT)

**Competing interests:** The authors declare that no competing interests exist.

## Introduction

Viruses are obligatory intracellular parasites that rely on the biochemical functions of their hosts to carry out infection. Although usually studied at the level of cell populations, viral infection is inherently a single-cell problem, where the outcome of infection can dramatically differ between genetically identical cells. For example, early studies in the 1940 s investigated the burst size of individually infected bacteria and concluded that it both spans three orders of magnitude and cannot be solely attributed to differences in bacteria size (*Delbrück, 1945*). A later study measured the burst size from individual HeLa cells infected with Herpes Simplex virus 1 (HSV-1) and found that many of the infected cells did not release viral progeny, that the variability between individual cells was high and that it did not correlate with the multiplicity of infection (MOI) used (*Wildy et al., 1959*). More recently, technological improvements have allowed the quantification of burst sizes and determination of the infection kinetics of different mammalian viruses, pointing to a high degree of cell-to-cell variability in infection (*Zhu et al., 2009*; *Timm and Yin, 2012*; *Schulte and Andino, 2014*; *Combe et al., 2015*; *Heldt et al., 2015*; *Cohen and Kobiler, 2016*; *Guo et al., 2017*; *Drayman et al., 2017*). One well-known source of this variability is the random distribution of the number of viruses that individual cells encounter (*Parker, 1938*; *Smith, 1968*; *Cohen and Kobiler, 2016*). Another source is genetic variability in the virus population, with some virus particles being

**eLife digest** Herpes simplex virus 1, or HSV-1, is a virus that infects most of the human population. In many people, the virus stays dormant in nerve cells, but in some individuals, it can 'wake up' regularly and cause painful facial lesions known as cold sores. In very few cases, the virus can enter the brain and become life threatening.

When HSV-1 encounters a human cell, there are three possible outcomes. The virus can either enter the cell and then replicate uncontrollably, get inside the cell but not multiply, or fail to enter the cell altogether. However, during experiments, researchers do not usually look at individual cells but instead consider whole populations. This makes it hard to understand the exact mechanisms that contribute to a cell resisting or succumbing to the virus.

New approaches are now making it possible to study individual cells over time. Here, Drayman et al. harnessed these methods to understand how individual human cells respond to HSV-1. The experiments show that most cells are actually able to resist the infection. Amongst those, a small fraction managed to stop the virus replicating by initiating a built-in 'antivirus program'. However, a minority of cells did become highly infected, shutting down the signaling process that fends off the virus. In these cells a different set of genes were switched on, making them more similar to the cells found in embryos. In the process, the virus recruited a protein called β-catenin to help with its multiplication.

There are efforts to develop drugs to interfere with β-catenin, as this protein is also produced differently in people with cancer. Such drugs, if identified and safe in humans, could potentially serve to treat HSV-1 infections.

DOI: https://doi.org/10.7554/eLife.46339.002

unable or less fit to establish infection (*Huang and Baltimore, 1970*; *Lauring et al., 2013*; *Stern et al., 2014*).

It is becoming clear that a third source of variability among individually infected cells is the host cell state at the time of infection (*Snijder et al., 2009*; *Snijder et al., 2012*; *Drayman et al., 2017*). Variability in the host cell state can arise from both deterministic processes, such as the cell-cycle, and stochastic processes, such as mRNA transcription and protein translation (*Elowitz et al., 2002*; *Cohen et al., 2008*; *Tay et al., 2010*; *Loewer and Lahav, 2011*; *Kellogg and Tay, 2015*). Recently, the advent of single-cell RNA-sequencing (scRNA-seq) has allowed researchers to examine virus–host interactions in multiple systems, mainly those of RNA viruses, highlighting the extreme cell-to-cell variability during viral infection (*Steuerman et al., 2018*; *Russell et al., 2018a*; *Xin et al., 2018*; *Zanini et al., 2018*; *Shnayder et al., 2018*). Thus, it is clear that a better understanding of viral infection requires studies at the single-cell level. Although scRNA-seq is providing a wealth of new information, it is currently limited to the characterization of highly abundant transcripts and can be augmented by other approaches such as live-cell imaging and RNA-sequencing of defined cell populations.

HSV-1 is a common human pathogen that belongs to the *herpesviridae* family and serves as the prototypic virus for studying alpha herpesviruses infection. De novo HSV-1 infection has both lytic and latent phases. In the lytic phase, the virus infects epithelial cells at the site of contact, replicates withing these host cells, before destroying them and releasing viral progeny. The latent phase is restricted to neurons, in which the virus remains silent throughout the host life with occasional reactivation. Here, we focus on the lytic part of the virus life cycle. Lytic infection is usually asymptomatic, but in some cases (particularly in immune-compromised individuals and infants), it can result in life-threatening conditions such as meningitis and encephalitis.

To initiate infection, HSV-1 must bind to its receptors, enter the cytoplasm, travel to the nuclear pore and inject its linear double-stranded DNA into the host nucleus (*Kobiler et al., 2012*). Once in the nucleus, viral gene expression proceeds in a temporal cascade involving three classes of viral genes: immediate-early (IE), early (E) and late (L) (*Honess and Roizman, 1974*; *Honess and Roizman, 1975*; *Harkness et al., 2014*). Transcription of IE genes is initiated by VP16 (*Weir, 2001*), whereas transcription of the E and L genes is activated by the IE protein ICP4 (*Dixon and Schaffer, 1980*; *Watson and Clements, 1980*; *DeLuca et al., 1985*). ICP0, another IE

gene, is a multifunctional E3-ubiquitn ligase that counteracts some of the host anti-viral systems (*Lanfranca et al., 2014*) and interferes with the action of transcriptional repressors (*Gu et al., 2005*; *Lomonte et al., 2004*; *Lutz et al., 2017*). Viral mutants that lack ICP0 expression (ΔICP0) are highly attenuated in a variety of cell types (*Stow and Stow, 1986*). Viral DNA replication occurs in sub-nuclear structures, called replication compartments (RCs), that aggregate the seven essential replication proteins as well as other viral and host proteins (*de Bruyn Kops and Knipe, 1988*; *Liptak et al., 1996*; *Weller and Coen, 2012*; *Dembowski and DeLuca, 2015*; *Dembowski et al., 2017*; *Reyes et al., 2017*; *Dembowski and DeLuca, 2018*). Upon viral DNA replication, ICP4 is predominantly localized in the RCs, with some diffuse nuclear and cytoplasmic localization (*Knipe et al., 1987*; *Zhu and Schaffer, 1995*).

Several studies applied high-throughput technologies to analyze the cellular response to HSV-1 infection at the population level. RNA sequencing revealed a widespread deregulation of host transcription, including the disruption of transcription termination (*Rutkowski et al., 2015*; *Hennig et al., 2018*), activation of anti-sense transcription (*Wyler et al., 2017*), depletion of RNA-polymerase II from the majority of host genes (*Abrisch et al., 2015*; *Birkenheuer et al., 2018*) and changes in splicing and polyadenylation (*Hu et al., 2016*). Although most cellular genes are downregulated by infection, some genes have been reported to be upregulated, including some anti-viral genes and genes encoding host transcription factors (*Pasieka et al., 2006*; *Taddeo et al., 2002*; *Hu et al., 2016*). Proteomics studies have defined the different stages and protein complexes that are present during HSV-1 replication (*Dembowski and DeLuca, 2015*; *Suk and Knipe, 2015*; *Dembowski et al., 2017*; *Reyes et al., 2017*; *Dembowski and DeLuca, 2018*), as well as the cellular protein response to infection (*Kulej et al., 2017*; *Lum et al., 2018*).

Although incredibly informative, population-level analyses suffer in that they average over all the cells in the population. In the case of virus-infected cells, the population is far from homogenous and could in fact contain opposite phenotypes, such as highly- infected and abortively infected cells, leading to contradictory results. One such example is the seemingly complex relation between HSV-1 infection and type I interferon (IFN) signaling. The picture that emerges from population-level measurements is paradoxical, with wildtype HSV-1 infection both clearly activating (*Gianni et al., 2013*; *Hu et al., 2016*; *Liu et al., 2016*; *Reinert et al., 2016*) and clearly repressing (*Lin et al., 2004*; *Johnson et al., 2008*; *Kew et al., 2013*; *Johnson and Knipe, 2010*; *Su et al., 2016*; *Christensen et al., 2016*; *Manivanh et al., 2017*; *Yuan et al., 2018*; *Chiang et al., 2018*) the type I IFN pathway. Such discrepancies might be resolved with the use of single-cell measurements.

Here, we apply a combination of live-cell time-lapse fluorescent imaging, scRNA-seq and the sequencing of sorted cell populations to explore HSV-1 infection at the single-cell level. We find that single cells that are infected by the virus show variability in all aspects of infection, starting from the initial phenotype (abortive infection vs. successful initiation of viral gene expression), through the timing and rate of viral gene expression, and ending with the host cellular response. This study resolves the apparent discrepancy in the literature regarding type I IFN induction and shows that this induction is restricted to a rare sub-population of abortively infected cells. Surprisingly, we find that the main transcriptional response in highly infected cells is the reprogramming of the cell to an embryonic-like state. We focus on the viral activation of the WNT/β-catenin pathway and find that β-catenin is recruited to the cell nucleus and the viral RCs, and is required for viral gene expression and progeny production.

## Results

### Viral infection dynamics vary among individual cells

We began by studying the temporal variability in viral gene expression initiation. To do so, we employed a wildtype HSV-1 (strain 17) that was genetically modified to express ICP4-YFP (*Everett et al., 2003*). Primary human fibroblasts (HDFn) were infected and monitored by time-lapse fluorescent microscopy (*Figure 1A*, *Figure 1—video 1*). HDFn were infected at an MOI of 2 (calculated on the basis of virus titration on Vero cells, which are ~2-fold more susceptible to HSV-1 infection than HDFn). This MOI was chosen because we found empirically that it resulted in ~50% of HDFn becoming ICP4-positive during primary infection. Note that we determined the genome:PFU

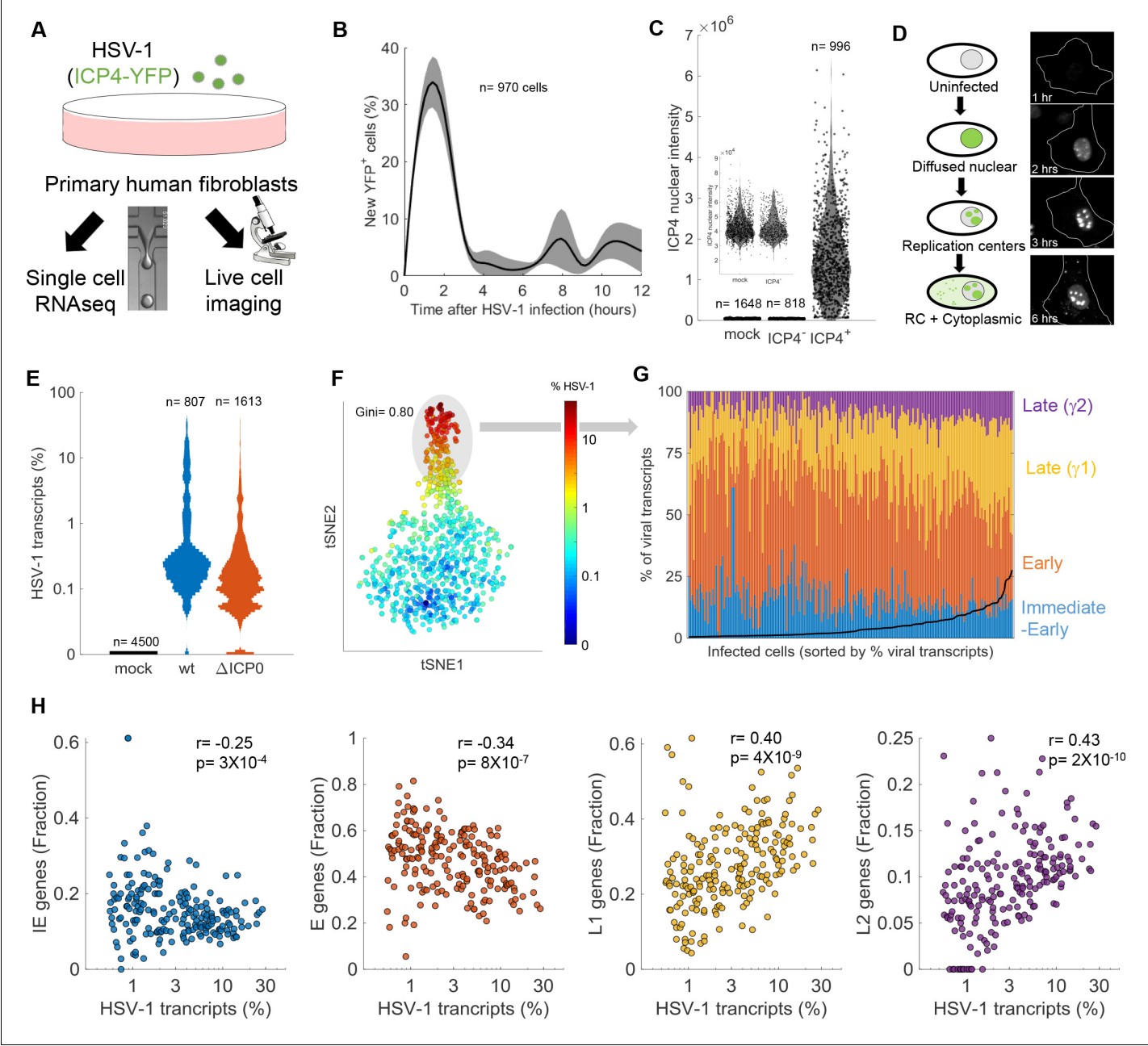

**Figure 1.** Cell-to-cell variability in infection dynamics and viral gene expression. (A) HDFn cells were infected with HSV-1 expressing ICP4-YFP and analyzed by time-lapse fluorescent imaging and scRNA-seq. (B) Distribution of the initial time of ICP4 expression. The black line denotes the average and the gray shadowing denotes the standard error of the time of ICP4 expression by single cells. n = 970 cells from three fields of view. (C) Violin plots showing the distribution of ICP4 nuclear intensity at 5 hr post infection of mock (n = 1648) or HSV-1 infected cells, either ICP4– (n = 818) or ICP4+ (n = 996). The smaller inset shows a zoomed-in view of the mock and ICP4– cells. Values of single cells are shown as circles. (D) Schematic diagram of the different localization phenotypes of ICP4 (left) and a representative single cell showing these phenotypes at different time points following infection. (E) Violin plots showing the distribution of the % of viral transcripts (out of the total transcripts) for individual cells that were mock-infected (n = 4500), infected with wt (n = 807) or infected with ΔICP0 (n = 1613) HSV-1. The y-axis is logarithmic. (F) T-distributed stochastic neighbor embedding (tSNE) plot based on viral gene expression in wt HSV-1 infection. Each dot represents a single cell and is colored according to the % of viral transcripts from blue (low) to red (high). The color bar is logarithmic. Gini is the Gini coefficient. (G) The relative abundance of the four viral gene classes: immediate-early (blue), early (orange), late γ1 (yellow) and late γ2 (purple) in highly infected single cells (highlighted by the gray oval in panel (F)). Single cells are ordered by their % of viral transcripts from low (left) to high (right), which are denoted by the black line. (H) Scatter plots of single cells showing the % of viral transcripts on the x-axis and the relative abundance of each viral gene class on the Y-axis. r and p are the Pearson correlation coefficients and p-values, respectively.

*Figure 1 continued*

DOI: https://doi.org/10.7554/eLife.46339.003

The following video and figure supplements are available for figure 1:

**Figure supplement 1.** Technical data relating to single-cell RNA-sequencing.

DOI: https://doi.org/10.7554/eLife.46339.004

**Figure supplement 2.** Joint analysis of host +viral genes in mock and wt-infected HDFn.

DOI: https://doi.org/10.7554/eLife.46339.005

**Figure supplement 3.** Cell-to-cell variability in viral gene expression upon ΔICP0 infection.

DOI: https://doi.org/10.7554/eLife.46339.006

**Figure 1—video 1.** Live imaging of HDFn infected by wildtype HSV-1 expressing ICP4-YFP.

DOI: https://doi.org/10.7554/eLife.46339.007

ratio for our viral stock and found it to be 36 ± 4, suggesting that all the cells in the culture have probably encountered numerous virus particles.

Initiation of ICP4 expression was observed to occur mostly between 1 hr and 4 hr post-infection (*Figure 1B*). Almost no new infections were observed between 4 hr and 6 hr post-infection, but two infection peaks were later seen at 8 and 11 hr. These peaks are probably the result of secondary infections, as new viral progeny can be detected in infected cells starting at 6 hr post-infection (*Pomeranz and Blaho, 2000*; *Ikeda et al., 2011*; *Drayman et al., 2017*). Given that the majority of infected cells have initiated viral gene expression by 5 hr, we chose this time point for further analyses. HDFn were infected as described above, fixed and stained with DAPI at 5 hr post-infection (to allow automated cell segmentation and quantification). This allowed the distinction of two cellular populations: cells that successfully initiated viral gene expression (ICP4$^+$) and cells that did not (ICP4$^-$). Of 1814 cells exposed to HSV-1, 996 cells (55%) were ICP4$^+$ and 818 (45%) were ICP4$^-$. Cells were classified as ICP4-negative or -positive on the basis of a threshold calculated from mock-infected cells (mean +3 standard deviations).

Among the ICP4$^+$ cells, nuclear levels of ICP4 varied by ~100 fold, ranging from $7 \times 10^4$ to $9 \times 10^6$ AU (*Figure 1C*). Infected cells showed three distinct phenotypes related to ICP4 localization (*Figure 1D*). Upon its expression, ICP4 is initially diffuse throughout the cell nucleus. As its level increases, ICP4 forms discrete foci in the nucleus. These are the viral RCs, where viral DNA replication takes place. Later, the levels of cytoplasmic ICP4 increase and interspersed foci can be seen in the cytoplasm. These phenotypes are temporally linked and delineate the progression through infection. As evident by time-lapse microscopy (*Figure 1—Video 1*), individual cells show a high degree of variability, not only in the timing of initial gene expression but also in the rate of infection progression.

Taken together, we find that not all cells that are exposed to HSV-1 successfully initiate viral gene expression under these experimental conditions. Those that do initiate viral gene expression show variation in the timing of initial gene expression, in the rate of infection progression and in the level and localization of the immediate-early protein ICP4. These results prompted us to explore cellular heterogeneity on a larger scale by applying single-cell RNA-sequencing (scRNA-seq) to infected cells.

## Viral gene expression is extremely variable among individual cells

HDFn were mock-infected or infected with wildtype HSV-1 (MOI 2, equivalent to ~70 genomes per cell, see 'Materials and methods') or a ΔICP0 HSV-1 mutant (MOI 0.5, equivalent to ~700 genomes per cell) and harvested for scRNA-seq at 5 hr post-infection. We chose to include the ΔICP0 mutant because it results in a relatively high number of abortive infections and in a robust activation of antiviral responses. For scRNA-seq, we applied the Drop-seq protocol (see 'Materials and methods' and *Macosko et al., 2015*). Briefly, a microfluidic device was used to encapsulate individual cells in a water-in-oil droplet in which cell lysis, mRNA-capture and barcoding took place. The barcoded mRNA was then recovered from the droplets, reverse-transcribed, amplified and sequenced. As each cDNA was barcoded with a cell and transcript ID, the sequencing data allow the reliable quantification of the number of transcripts in individual cells. Technical data on sequencing depth and filtering criteria are presented in *Figure 1—figure supplement 1*.

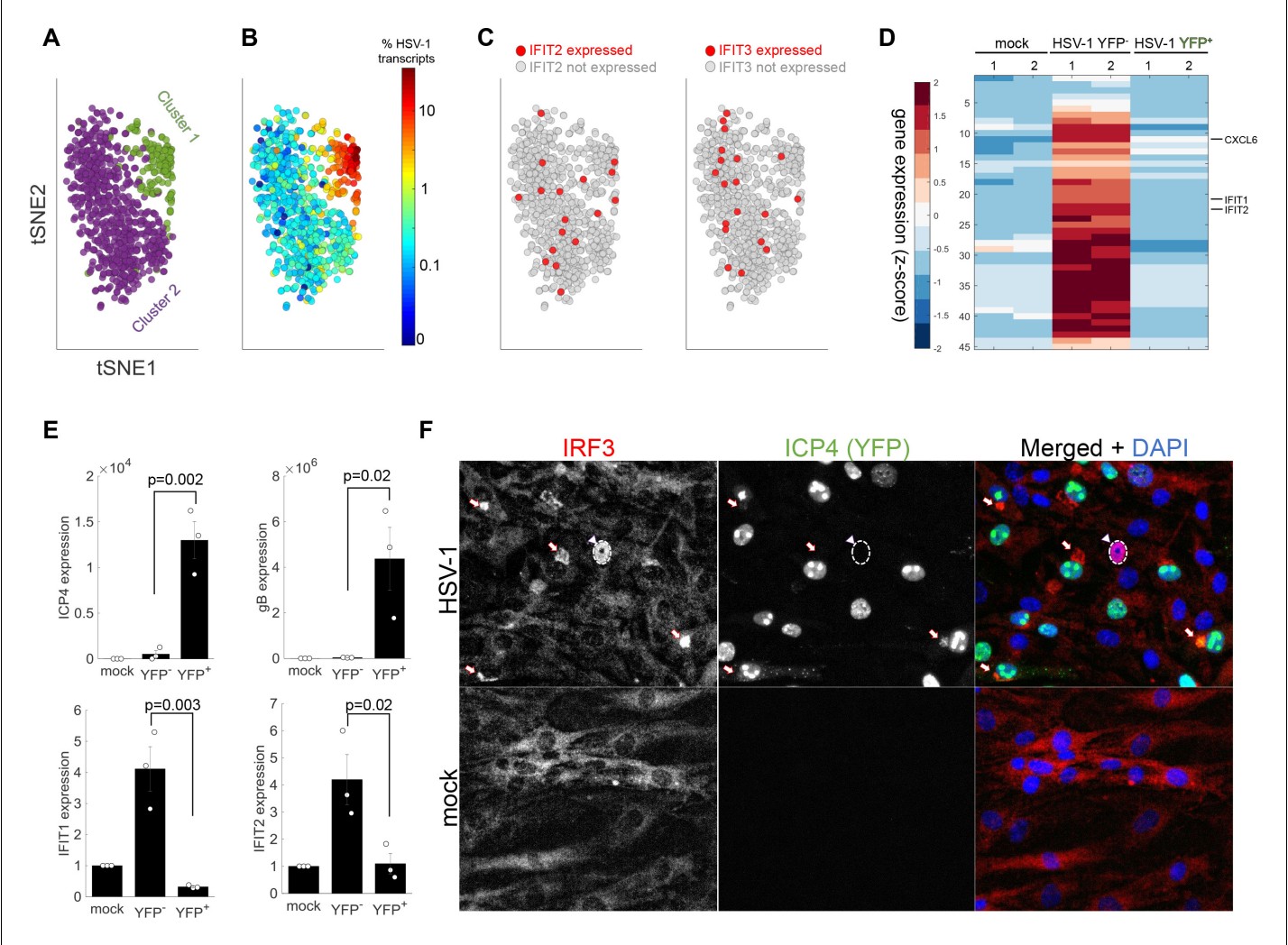

**Figure 2.** The anti-viral program is initiated in a rare sub-population of abortively infected cells. (**A**) tSNE plot based on viral and host gene expression. Cells are colored according to their clustering. Cluster one is colored green and cluster two is colored purple. (**B**) tSNE plot as in panel (A). Cells are colored according to their % of viral transcript expression. (**C**) tSNE plot as in panel (A). Cells are colored according to their expression (red) or lack of expression (gray) of *IFIT2* (left) or *IFIT3* (right). (**D**) Heat-map of genes that are significantly upregulated in ICP4⁻ cells, as compared to both mock and ICP4⁺ cells. RNA-sequencing was performed in duplicates denoted by the numbers 1 and 2 on the top row. Each row shows the normalized (z-score) expression of a single gene, colored from low (blue) to high (red). (**E**) Validation of selected genes by quantitative PCR. Bar plots show the expression levels of the viral genes *ICP4* and *gB* (top) and the anti-viral genes *IFIT1* and *IFIT2* (bottom). Values are mean ± s.e. of three independent biological repeats. Individual measurements are shown as circles. p-values were calculated using a one-tailed t-test. (**F**) Immunoflorescent staining of IRF3 in mock-infected (bottom) or HSV-1 infected (top) cells at 5 hr post infection. The arrowhead points to an ICP4-negative cell that shows nuclear IRF3 staining (nucleus border denoted by a dashed white line). The arrows point to aggregations of IRF3 outside the nucleus in ICP4-positive cells.

DOI: https://doi.org/10.7554/eLife.46339.008

The following figure supplements are available for figure 2:

**Figure supplement 1.** Cell-cycle is anti-correlated with viral gene expression and is a major source of transcriptional variability.
DOI: https://doi.org/10.7554/eLife.46339.009
**Figure supplement 2.** ISGs expression in single-cells infected by wildtype HSV-1.
DOI: https://doi.org/10.7554/eLife.46339.010

Only 0.4% of mock-infected cells had any reads aligned to the HSV-1 genome, with a maximal expression of two viral gene counts (0.05% of transcripts). Cells that were infected with either wt or ΔICP0 HSV-1 showed extreme cell-to-cell variability in the amount of viral transcripts that they expressed, ranging from 0–36% (*Figure 1E*). The viral gene expression distribution was highly skewed, with most cells expressing low levels of viral transcripts and some cells expressing much

higher levels (*Figure 1E*). A joint analysis of mock and wt-infected cells showed that highly infected cells clustered separately from the mock-infected and low-infected cells, which were intermingled (*Figure 1—figure supplement 2*).

The Gini coefficient, a measurement of population inequality ranging from zero (complete equality) to one (complete inequality), was used to evaluate the distribution of viral gene expression among individual cells. The Gini coefficients were 0.8 for wildtype infection and 0.77 for ΔICP0 infection, higher than that reported for viral gene expression by an influenza virus (0.64; *Russell et al., 2018a*). When wildtype viral gene expression is visualized in two-dimensions (using the tSNE dimensionality reduction technique; *van der Maaten and Hinton, 2008*), two clusters of cells can be seen, which are distinguished by the amount of viral gene expression (less or more than ~1%, *Figure 1F*). A similar distribution was seen for ΔICP0-infected cells, although there were significantly fewer cells in the 'highly infected' cluster in this case (*Figure 1—figure supplement 3*).

To further explore cell-to-cell variability in viral gene expression, we analyzed the relative expression of the four temporal groups of viral transcripts: immediate-early (IE), early (E) and late (subdivided into early-late (γ1) and true-late (γ2)). We focused on the group of highly infected cells, because the lowly infected cells had too few viral gene counts for accurate analysis. *Figure 1G* shows the relative expression of the viral gene classes in single cells, ordered from low to high viral gene expression. The fraction of late genes increases as total viral gene expression increases, at the expense of IE and E genes. The correlations between viral gene expression and the four classes of viral transcripts are shown in *Figure 1H–K*. Similar observations were made for ΔICP0-infected cells (*Figure 1—figure supplement 3*).

Our scRNA-seq data indicate a wide and uneven distribution of viral gene expression during HSV-1 infection, with most cells expressing no or low levels of viral gene transcripts and a smaller group expressing much higher levels (in agreement with the ICP4 expression levels presented above). The vast majority of cells exposed to HSV-1, either wildtype or ΔICP0, had some level of viral gene expression, suggesting that the fraction of lowly expressing cells (and the ICP4⁻ population noted above) are indeed abortively infected cells, rather than cells that did not encounter a virus. We note that significant cell-to-cell differences are seen even within the group of highly infected cells, with viral gene expression ranging from 1% to >30%, and that this 'viral expression load' is correlated with late gene expression.

## The cell-cycle affects HSV-1 gene expression

We have previously shown that the cell-cycle stage at the time of infection is a cellular determinant of successful HSV-1 infection in the H1299 cell line (*Drayman et al., 2017*). In that study, we found that cells that were infected in the $G_2$ phase were less likely to initiate viral gene expression and that cellular escape from viral-enforced mitotic arrest is highly detrimental to HSV-1 infection. To evaluate the effect of the cell-cycle in HDFn infection, we calculated a cell-cycle score for each cell in our dataset (see 'Materials and methods' section and *Tirosh et al., 2016*) and measured the correlation between HSV-1 gene expression and the cell-cycle score (*Figure 2—figure supplement 1*). We found that viral gene expression was negatively correlated with the cell-cycle score, with cells in the later parts of the cell-cycle expressing ~10-fold fewer viral genes than those in the early part of the cycle, in agreement with our previous finding in the H1299 cell-line.

As the cell-cycle is both a major source of cell-to-cell variability and negatively correlated with viral gene expression, it was crucial to regress out the cell-cycle effect before analyzing the host response (*Figure 2—figure supplement 1*). We could now turn to analyzing the host genes that are differentially expressed among HSV-1-infected cells, starting with the anti-viral response.

## The anti-viral program is initiated by a rare sub-population of abortively infected cells

Previous population-level studies reported the activation of anti-viral genes during wildtype HSV-1 infection, so we hypothesized that highly infected cells (*Figure 2A,B*, cluster 1) should be enriched for anti-viral genes. To our surprise, differential gene expression analysis of the two clusters did not indicate upregulation of the anti-viral response in cluster 1 (*Supplementary file 1a*). In fact, canonical anti-viral genes such as *IFIT2* and *IFIT3* were only detected in 2–3% of the cells from both clusters 1 and 2 (*Figure 2C*). When comparing a larger panel of interferon-stimulated genes (ISGs) in high- vs

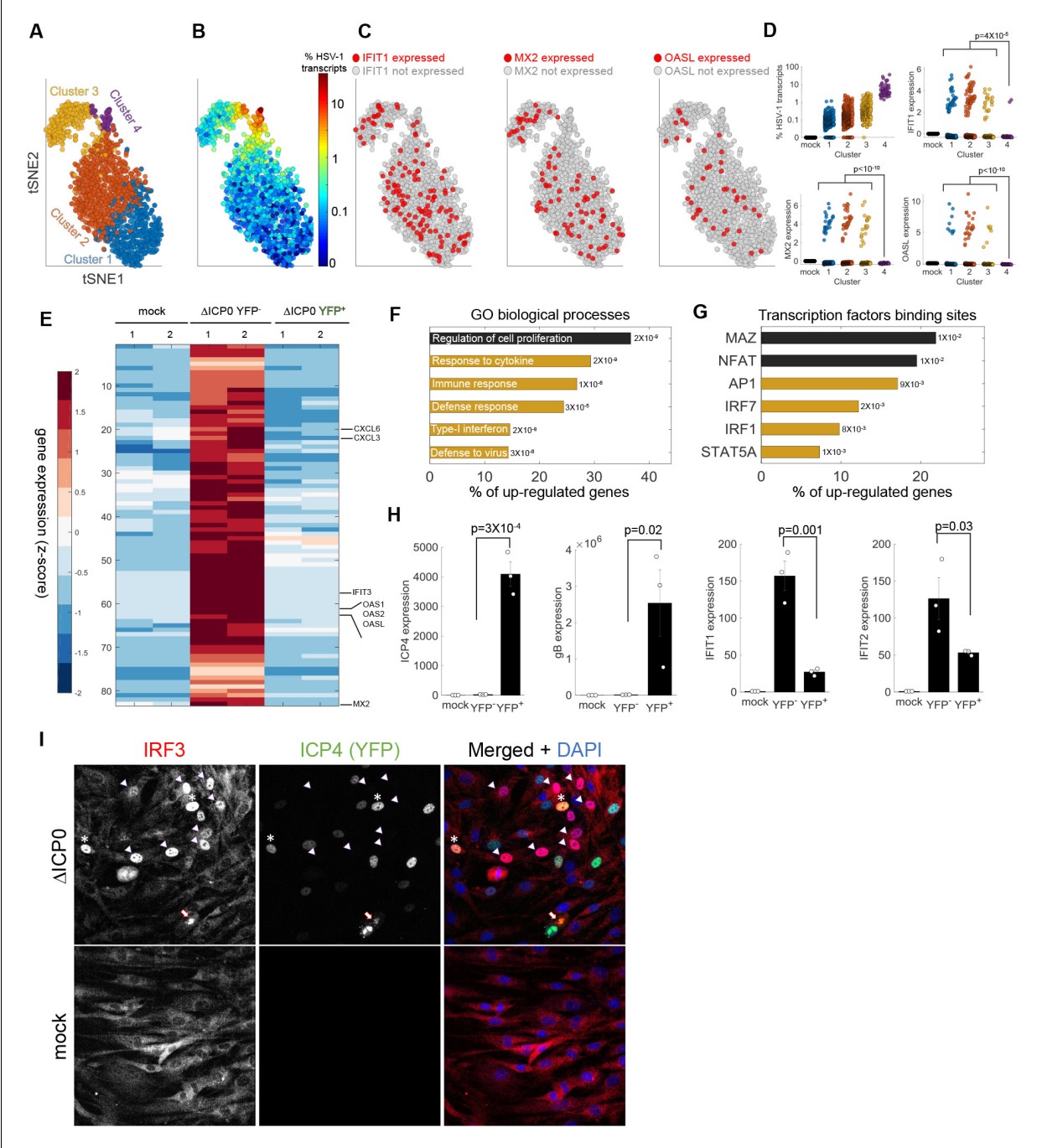

**Figure 3.** Anti-viral signaling in cells infected by ΔICP0 mutant. (**A**) tSNE plot based on viral and host gene expression. Cells are colored according to their clustering. (**B**) tSNE plot as in panel (A), with cells colored according to the % expressed transcripts that are viral. (**C**) tSNE plots as in panel (A). Cells are colored according to expression (red) or lack of expression (gray) of *IFIT1* (left), *MX2* (middle) or *OASL* (right). (**D**) Scatter plots showing the expression of HSV-1 transcripts (top left) and the expression of the anti-viral genes *IFIT1*, *MX2* and *OASL* in mock-infected cells (first column) and in each cluster of cells infected by ΔICP0. p-values were calculated by a Chi-square test, comparing cluster 4 to clusters 1–3. (**E**) Heat-map of genes that are significantly upregulated in ΔICP0-infected ICP4⁻ cells, as compared to both mock and ICP4⁺ cells. RNA-sequencing was performed in duplicates, as denoted by the numbers 1 and 2 on the top row. Each row shows the normalized (z-score) expression of a single gene, colored from low (blue) to high (red). (**F**) Gene Ontology (GO) terms for the enriched genes identified in panel (E). The term is written on the bar, and the value next to each bar is the FDR-corrected p-value for its enrichment. Bar length shows the % of upregulated genes annotated to the GO term. Highlighted in yellow are GO terms that are associated with anti-viral signaling. (**G**) Transcription factor binding sites that are enriched in the promoters of genes identified in panel (E). (**H**) QPCR validation of selected genes. The bar plots show the expression level of the viral genes *ICP4* and *gB* and of the anti-viral genes *IFIT1* and *IFIT2*. Values are means ± s.e. of three independent biological repeats. Individual measurements are shown as circles. p-values

*Figure 3 continued on next page*

*Figure 3 continued*

were calculated using a one-tailed t-test. (I) Immunoflorescent staining of IRF3 in mock-infected (bottom) or ΔICP0-infected (top) cells at 5 hr post infection. The arrowheads point to ICP4-negative cells that contain nuclear IRF3. Asterisks denote ICP4-positive cells that contain nuclear IRF3. The arrow points to a cell in the late stages of infection, in which IRF3 is aggregated in the nuclear periphery.

DOI: https://doi.org/10.7554/eLife.46339.011

The following figure supplement is available for figure 3:

**Figure supplement 1.** ISGs expression in single-cells infected by ΔICP0 HSV-1.
DOI: https://doi.org/10.7554/eLife.46339.012

low-infected cells (*Figure 2—figure supplement 2*), most ISGs are in fact more highly expressed in cells with low HSV-1 gene expression.

One possible explanation is that anti-viral genes are indeed expressed in highly infected cells but were not detected by scRNA-seq because of technical limitations. To investigate this, infected cells were sorted by fluorescence-activated cell sorting (FACS) into two populations based on ICP4-YFP expression (ICP4$^+$ and ICP4$^-$), and each population was sequenced. In agreement with the scRNA-seq data, the expression of canonical anti-viral genes was not significantly different between mock-infected and ICP4$^+$ cells. Rather, our analysis indicated that a small group of genes, including the anti-viral genes *IFIT1* and *IFIT2*, were specifically upregulated in the ICP4$^-$ population (*Figure 2D*, *Supplementary file 2a*). The Gene Ontology (GO) biological processes associated with these upregulated genes included terms such as 'response to type I interferon' and 'immune response' (*Supplementary file 2b*). Validation of selected transcripts by quantitative PCR (QPCR) is shown in *Figure 2E*.

ISGs are usually expressed after interferon stimulation, but we failed to detect the activation of any of the interferon genes in either the scRNA-seq or the sequencing of the sorted cell populations. To pinpoint the origin of the anti-viral response, cells were stained for IRF3, because IRF3 translocation from the cytoplasm to the nucleus is one of the first steps in type I interferon induction. In ICP4$^+$ cells, IRF3 was blocked from entering the nucleus and concentrated in the nuclear periphery (*Figure 2F*), while a rare subset of ICP4$^-$ cells (<1%) showed nuclear localization of IRF3. We thus conclude that wildtype HSV-1 infection efficiently blocks the induction of the anti-viral response and that the activation of anti-viral genes in population-averaged measurements is the result of anti-viral signaling that is elicited in a rare population of abortively infected cells.

We next evaluated the anti-viral response in cells infected by ΔICP0 HSV-1. ICP0 is a multifunctional viral protein, which blocks IRF3 signaling (*Lin et al., 2004*). Cells infected with ΔICP0 clustered into four groups (*Figure 3A*). Cluster 1 consists of cells with very few viral transcripts. Clusters 2 and 3 have slightly higher viral gene expression, and the small cluster 4 consists of highly infected cells (*Figure 3B,D*). Although the magnitude of the anti-viral response in ΔICP0-infected cells was much greater than that in the wildtype-infected cells (*Figure 2*), it was still only observed in a small population of cells, with ~8% of the cells expressing *IFIT1*, *MX2* or *OASL* (compared to none of the mock-infected cells). These cells had low viral gene expression levels and mostly belonged to clusters 1–3 (*Figure 3C,D*). Anti-viral signaling was not seen in the highly infected cells of cluster 4, with the exception of two cells expressing *IFIT1* (*Figure 3C,D*). As described above for the wildtype infected cells, we also compared a larger panel of ISGs in highly vs lowly infected cells (*Figure 3—figure supplement 1*), with similar results.

RNA-sequencing of sorted cells that were infected with ΔICP0 identified ~80 genes that are significantly upregulated in ICP4$^-$ cells compared to mock and ICP4$^+$ cells (*Figure 3E*, *Supplementary file 2c*). These genes were enriched for functional annotations of anti-viral signaling (*Figure 3F*, *Supplementary file 2d*) and for binding sites of the transcription factors IRF1, IRF7 and STAT5 (*Figure 3G*, *Supplementary file 2e*). An important difference from wildtype infection is that, although enriched in the ICP4$^-$ population, these anti-viral genes are also activated in the ICP4$^+$ population, albeit to a lesser extent (*Figure 3H*). We confirmed this observation through immunofluorescent staining of IRF3 (*Figure 3I*). The staining showed that a higher proportion of ΔICP0-infected cells showed nuclear IRF3 localization when compared to wildtype-infected cells. The majority of cells with nuclear IRF3 were ICP4$^-$ but some ICP4$^+$ cells also showed nuclear IRF3 staining. These ICP4$^+$ cells showed diffuse nuclear localization of ICP4, indicating that their infection was aborted prior to the generation of replication compartments (*Figure 1D*). The few cells that were

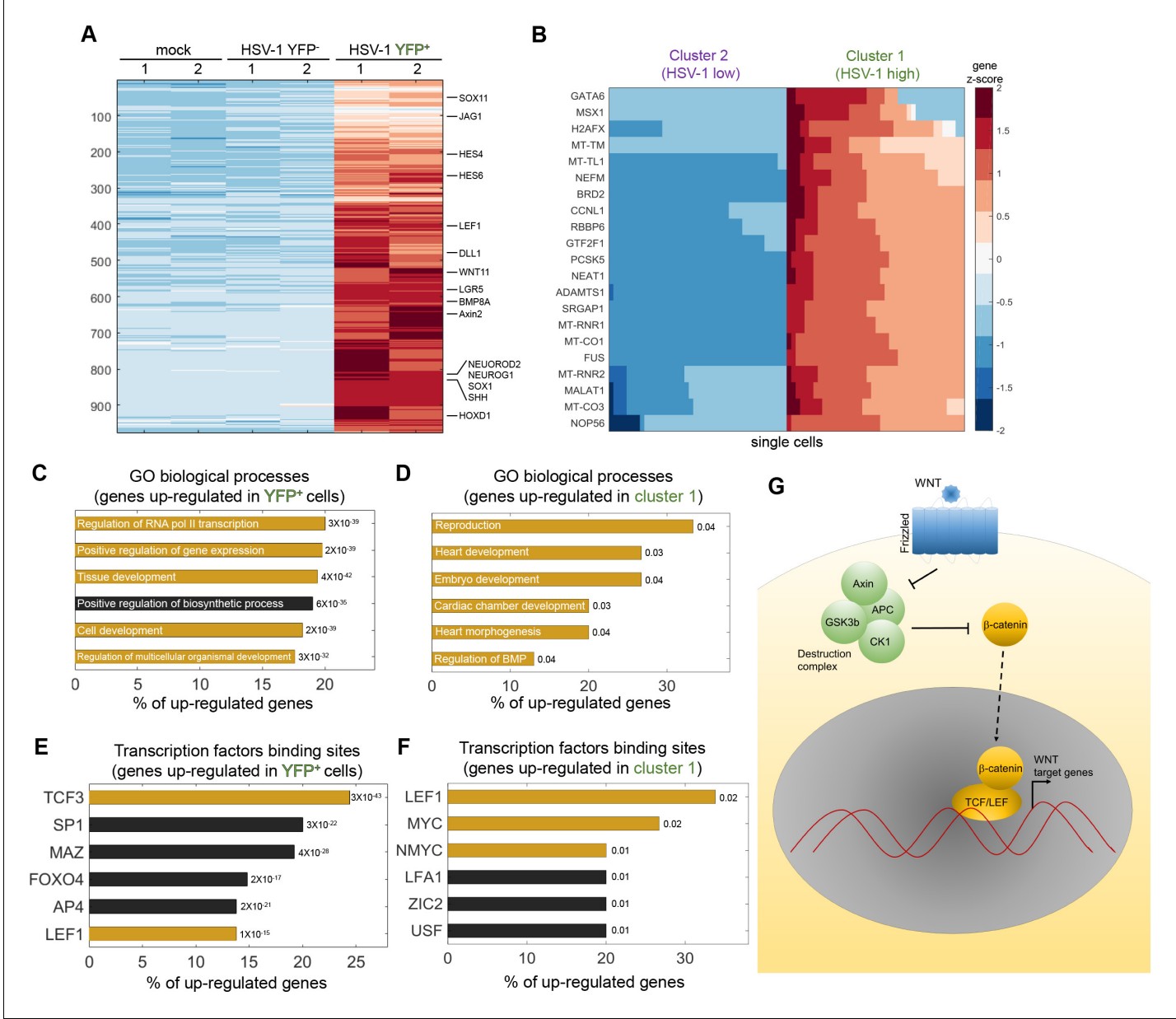

**Figure 4.** HSV-1 infection upregulates developmental pathways. (**A**) Heat-map of genes that are significantly upregulated in ICP4+ cells, as compared to both mock-infected and ICP4− cells. RNA-sequencing was performed in duplicates denoted by the numbers 1 and 2 on the top row. Each row shows the normalized (z-score) expression of a single gene, colored from low expression (blue) to high expression (red). (**B**) Heat-map of genes that are significantly upregulated in cluster 1 (highly infected) compared to cluster 2 (lowly infected) single cells. Each row shows the normalized (z-score) expression of a single gene, colored from low expression (blue) to high expression (red) in 80 single cells (40 from cluster 1 and 40 from cluster 2). The color bar is shared for both panels (A) and (B). (**C,D**) GO terms enriched for genes identified in panels (A) and (B), respectively. The GO term is written on the bar and the value next to each bar is the FDR-corrected p-value for its enrichment. Bar length shows the % of upregulated genes annotated to the GO term. GO terms that are associated with development and gene regulation are highlighted in yellow. (**E,F**) Transcription-factor-binding sites that are enriched in the promoters of genes identified in panels (A) and (B), respectively. The value next to each bar is the FDR-corrected p-value for the enrichment of the corresponding transcription-factor-binding site. Bar length shows the % of upregulated genes that contain binding sites for each transcription factor. Highlighted in yellow are the LEF1, TCF3, MYC and NMYC transcription factors, which are part of the WNT/β-catenin pathway. (**G**) A simplified diagram of the WNT/ β-catenin signaling pathway that controls LEF/TCF transcriptional activity.

DOI: https://doi.org/10.7554/eLife.46339.013

The following figure supplement is available for figure 4:

**Figure supplement 1.** ΔICP0 infection upregulates developmental pathways.
DOI: https://doi.org/10.7554/eLife.46339.014

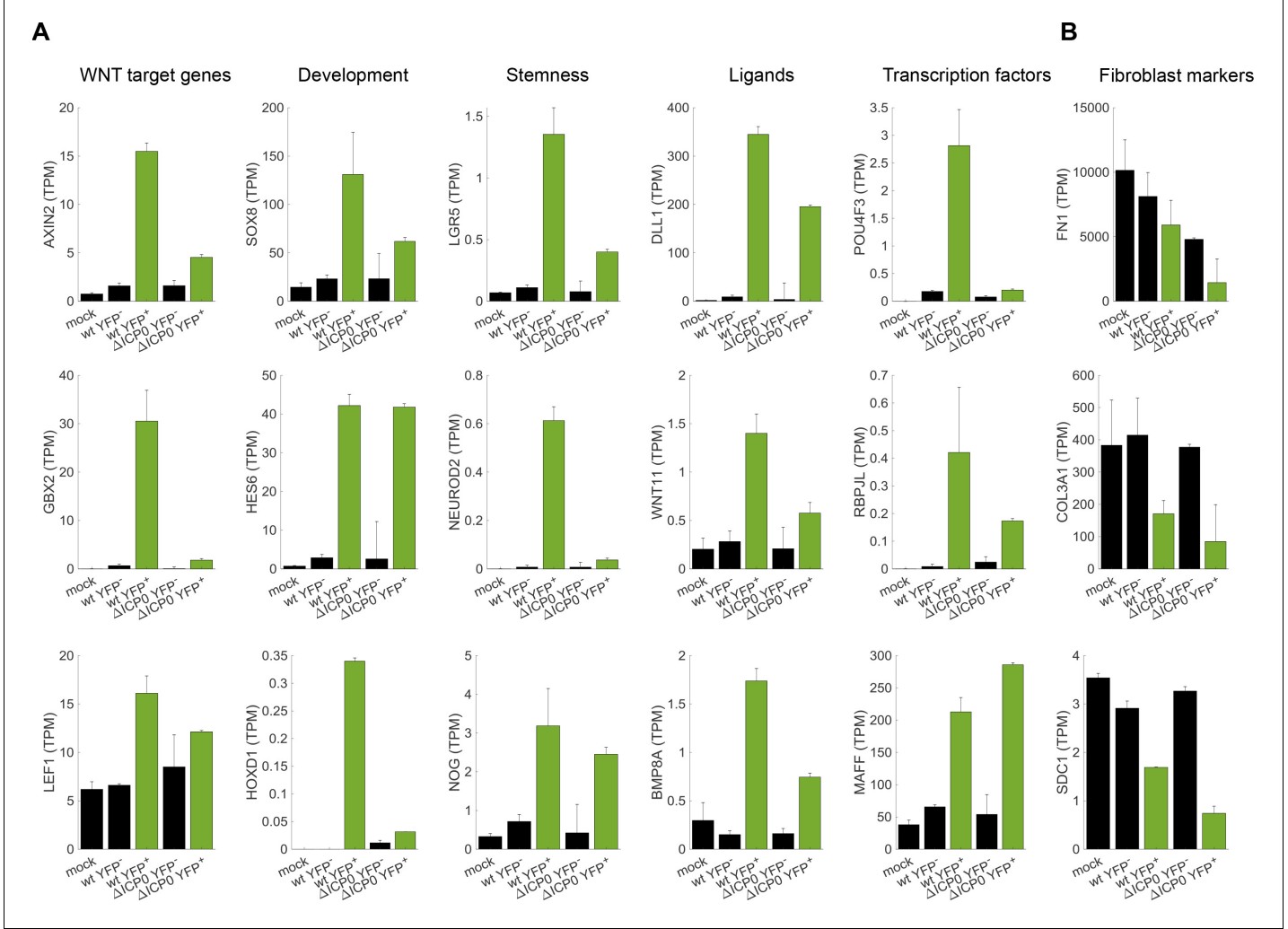

**Figure 5.** Cellular reprogramming during HSV-1 infection. (**A**) Bar plots showing the expression level (transcripts per million (TPM)) of selected examples of genes that participate in developmental pathways and are upregulated in HSV-1-infected cells. Black bars denote mock-infected and ICP4⁻ cells. Green bars denote ICP4⁺ cells. Values are means ±s.e of the two sequenced biological replicates (*Figure 4A* and *Figure 4—figure supplement 1A*). (**B**) Bar plots showing the expression level of selected examples of fibroblast marker genes.

DOI: https://doi.org/10.7554/eLife.46339.015

able to proceed to the later stages of infection (as indicated by the appearance of replication compartments and cytoplasmic ICP4 foci) showed the same peri-nuclear aggregation of IRF3 as wild-type-infected cells (*Figure 3I* and *Figure 2F*).

Altogether, the sequencing of both single cells and sorted cell populations, as well as immunofluorescence staining of IRF3, suggests that the anti-viral program is initiated in a small subset of abortively infected cells but is blocked in highly infected cells, even in the absence of ICP0. This behavior explains the apparent discrepancy in previous population-level measurements that showed both activation and inhibition of type I interferon signaling during HSV-1 infection.

## HSV-1 infection results in transcriptional reprogramming of the host to an embryonic-like state

We next focused on genes that are upregulated during HSV-1 infection, in either the scRNA-seq or sorted cell population experiments. A total of 977 genes were significantly upregulated in wildtype ICP4⁺ cells as compared to both ICP4⁻ and mock-infected cells in the bulk RNA-sequencing (*Figure 4A*, *Supplementary file 3a*). In the single-cell RNA-sequencing, 21 genes were significantly

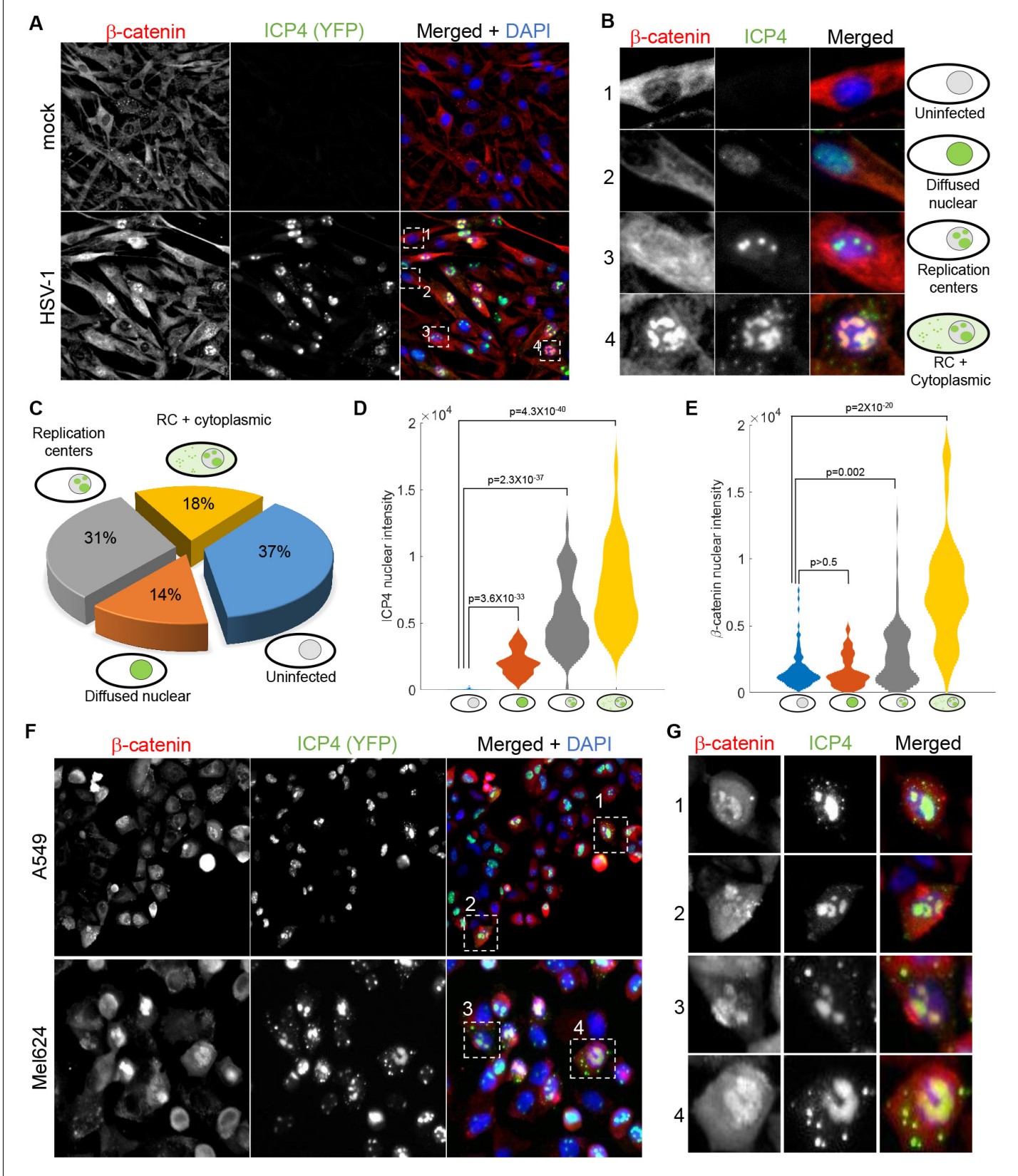

**Figure 6.** β-catenin translocates to the nucleus and concentrates in the viral replication compartments. (A) Immunoflorescent staining of β-catenin in mock-infected (top) or HSV-1 infected (bottom) HDFn cells at 5 hr post infection. (B) Magnified images of the four cells denoted by dashed white boxes

*Figure 6 continued on next page*

*Figure 6 continued*

in panel (A), showing representative images of cells with different ICP4 and β-catenin localizations. (C) Quantification of the relative abundances of the different ICP4 localizations (n = 204 cells). (D,E) Violin plots showing the distributions of ICP4 (D) and β-catenin (E) nuclear levels in cells showing the four ICP4 localization phenotypes. p-values were calculated by a two-tailed two-sample t-test and corrected for multiple comparisons by the Bonferroni correction. (F,G) Immunofluorsence as in panels (A) and (B) for A549 and Mel624 cells.

DOI: https://doi.org/10.7554/eLife.46339.016

The following figure supplement is available for figure 6:

**Figure supplement 1.** β-catenin translocates to the nucleus and concentrates in the viral replication compartments upon ΔICP0 infection.

DOI: https://doi.org/10.7554/eLife.46339.017

upregulated in highly infected single cells (*Figure 4B*, *Supplementary file 1a*). Remarkably, we found that a major portion of these upregulated genes are associated with GO terms that concern the regulation of RNA transcription and developmental processes (*Figure 4C,D* and *Supplementary files 1b* and *3b*). Similar results were observed in cells infected with ΔICP0 (*Figure 4—figure supplement 1* and *Supplementary files 1d-f* and *3d-f*).

The promoters of these upregulated genes are enriched for the binding sites of several transcription factors, including LEF1, TCF3 and MYC (*Figure 4E,F* and *Supplementary files 1c* and *3c*). 23% of the promoters of the upregulated genes in ICP4$^+$ cells contained a binding site for the TCF/LEF transcription factors, which are activated by the WNT/β-catenin pathway (*Sokol, 2011*) (*Figure 4G*). Note that *LEF1* itself is a WNT target gene that is upregulated during HSV-1 infection (*Figure 5A*).

Examples of upregulated genes are shown in *Figure 5A* and include canonical WNT target genes such as *AXIN2*, key developmental genes belonging to the *SOX*, *HOX* and *HES* families, stem-cell associated transcripts such as *LGR5*, and a multitude of extra-cellular ligands of various developmental pathways, including the WNT, Notch, Hedgehog and TGFβ signaling pathways. In agreement with the less-efficient infection by ΔICP0, most of these transcripts are also upregulated in ΔICP0-infected ICP4$^+$ cells, but to a lesser degree than in wildtype-infected cells. Concomitant with the establishment of this embryonic-like transcriptional program, we observed a reduction in the levels of key fibroblast-marker genes, such as those encoding α1(III) collagen and fibronectin (*Figure 5B*).

We conclude that cells that are highly infected by HSV-1 undergo de-repression of embryonic and developmental genetic programs, including the WNT/β-catenin pathway.

## β-catenin translocates to the nucleus and concentrates in the viral replication compartments

As many of the upregulated genes in infected cells are known WNT target genes and/or contain LEF/TCF-binding sites in their promoters, we investigated the state of β-catenin in these cells. Infected cells were fixed and stained for β-catenin at 5 hr post-infection (*Figure 6A*). As expected, β-catenin was mainly cytoplasmic/membrane-bound in mock-infected cells. In HSV-1-infected cells, β-catenin showed three distinct localization patterns: un-perturbed (cytoplasmic), diffuse nuclear or aggregated in nuclear foci (*Figure 6A,B*). Similar results were obtained for cells infected by ΔICP0 (*Figure 6—figure supplement 1*).

At 5 hr post-infection, 37% of the cells were ICP4 negative, 14% were at the earliest stage of infection (diffuse nuclear ICP4), 31% had assembled viral replication compartments and 18% had progressed to show cytoplasmic foci of ICP4 (*Figure 6C*). ICP4 levels increased from one group to the next, in accordance with the temporal progression of infection (*Figure 6D*).

β-catenin localization was linked to the temporal progression of infection. β-catenin remained cytoplasmic in both ICP4$^-$cells and cells with diffused nuclear ICP4, translocated to the nucleus upon the generation of the replication compartments and subsequently co-localized with the RCs, but only in cells showing cytoplasmic foci of ICP4 (*Figure 6B,E*). A similar phenotype was seen in two epithelial cell-lines: A549, a lung cancer cell-line, and Mel624, a patient-derived melanoma cell-line (*Figure 6F*).

This analysis indicates that β-catenin is indeed co-opted by HSV-1. Cell-to-cell variability in the progression of infection results in heterogeneity of β-catenin localization, with recruitment of β-catenin to the viral replication compartments occurring during the later stages of infection.

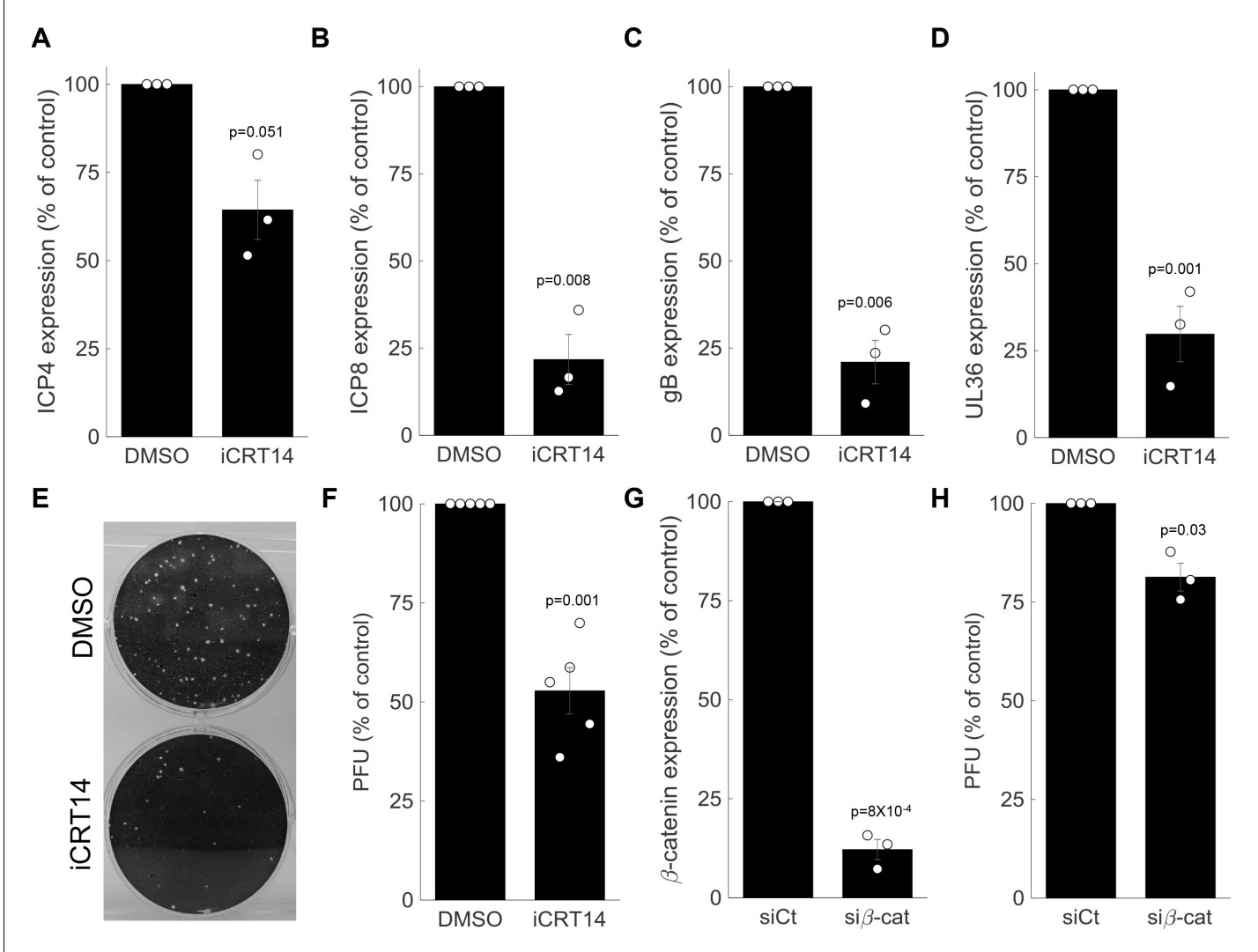

**Figure 7.** β-catenin is required for late viral gene expression and progeny production. (**A-D**) HDFn treated with the β-catenin inhibitor iCRT14 or with vehicle alone (dimethyl sulfoxide (DMSO)) were infected with HSV-1 and analyzed for viral gene expression by RT-PCR at 5 hr post infection. Results show the means ± s.e. for three biological repeats, as compared to the DMSO treatment. The genes analyzed were *ICP4* (A), *ICP8* (B), *gB* (C) and *UL36* (D). Circles are the results of individual experiments. p-values were calculated using a two-tailed one-sample t-test. (**E**) Representative image of a plaque assay, titrating virus generated from a 24-hr infection of HDFn treated with iCRT14 or DMSO. (**F**) Titers were calculated for viral progeny produced from iCRT14 or DMSO treated as in panel (E). The bars show the means ±s.e. from five biological repeats, as compared to the DMSO treatment. Circles are the results of individual experiments. p-values were calculated using a two-tailed one-sample t-test. (**G,H**) HDFn were treated with control short interfering RNA (siRNA) (siCt) or siRNA targeting β-catenin (siβ-cat) for 72 hr. Cells were assessed for β-catenin expression (G) or were infected for 24 hr before viruses were harvested and titrated as in panel (E) (H). Bars show the means ±s.e. from three biological repeats. p-values were calculated using a two-tailed one-sample t-test.

DOI: https://doi.org/10.7554/eLife.46339.018

## β-catenin activation is necessary for late viral gene expression and progeny production

As β-catenin target genes are activated by the virus, and because β-catenin is recruited to the viral replication centers, we hypothesized that β-catenin activity is required for the completion of the viral life cycle. We infected cells treated with an inhibitor of β-catenin activity, iCRT14 (*Gonsalves et al., 2011*), and measured viral gene expression at 5 hr post-infection (*Figure 7A–D*). Our results indicate that β-catenin inhibition had no or minimal impact on immediate-early gene expression (*Figure 7A*)

but significantly inhibited early and late gene expression (*Figure 7B–D*). These observations are in agreement with the late recruitment of β-catenin to the viral RCs described above.

To measure the impact of β-catenin inhibition on viral progeny formation, we treated the cells with iCRT14, infected the cells for 24 hr and harvested and titrated the resulting viral progeny by plaque assay (*Figure 7E*). In accordance with its impact on late viral gene expression, β-catenin inhibition significantly reduced viral progeny formation (*Figure 7F*). This is in agreement with a recent report by *Zhu and Jones (2018)* showing iCRT14-reduced plaque formation in two other cell types (HLF and Vero). Similar results were obtained when β-catenin was silenced using siRNA (*Figure 7G, H*).

Taken together, our data show that HSV-1 reprograms the cell to an embryonic-like state, in part through the co-option of β-catenin, which is needed for late viral gene expression and progeny production.

## Discussion

In this study, we applied a combination of time-lapse fluorescent imaging, scRNA-seq and sequencing of sorted cell populations to understand HSV-1 infection at the single-cell level. We find that single cells tha are infected by the virus show variability in all aspects of infection, starting from the initial phenotype (abortive infection vs. successful initiation of viral gene expression), through the timing and rate of viral gene expression and ending with the cellular response of the host cell. Such heterogeneity in the population of infected cells is detrimental when performing population-averaged measurements but can be untangled through single-cell analyses, which can provide new insights into biological processes.

With regard to IFN response, we find that two opposite phenotypes exist in the population of infected cells, explaining the discrepancy in the literature. Surprisingly, we find that IFN induction is limited to a small group of abortively infected cells, even in cells infected by the ΔICP0 mutant (*Figures 2* and *3*). This rare activation of anti-viral signaling seems to be widespread, as single-cell investigations into RNA viruses such as West-Nile virus (*O'Neal et al., 2019*) and influenza (*Russell et al., 2018a*; *Russell et al., 2018b*) have reported similar findings. Why only a subset of cells activate the anti-viral program is an intriguing question and several hypotheses come to mind. For example, these cells could be poised for IFN induction due to stochastic variability in expression of the signaling pathway components (*Zhao et al., 2012*; *Patil et al., 2015*) or IFN induction might be linked to the number of viral particles that a cell encounters. We plan to pursue and further characterize these rare cells in future studies.

We further found that highly infected cells undergo transcriptional reprogramming and activate multiple developmental pathways. This is exemplified by changes in β-catenin localization that correspond to distinct stages in HSV-1 infection (*Figure 6*): β-catenin is first recruited to the cell nucleus and then later to the viral replication compartments. These findings augment a growing body of literature that shows a link between viral infection and the β-catenin pathway (reviewed in *van Zuylen et al., 2016*), such as during infections with murine cytomegalovirus (MCMV) (*Juranic Lisnic et al., 2013*), influenza (*More et al., 2018*), hepatitis B virus (HBV) (*Daud et al., 2017*) and Rift Valley Fever virus (*Harmon et al., 2016*).

How HSV-1 infection causes this massive reprogramming of the host cell state is currently unknown, although we can rule out a direct involvement of ICP0 because this reprogramming also occurs during infection with ΔICP0, albeit to a lesser extent. β-catenin activation is certainly one part of this, but it is likely that the expression of epigenetic regulators is also important. Indeed, a proteomics study of host chromatin during HSV-1 infection has identified widespread changes in the epigenomic landscape of infected cells (*Kulej et al., 2017*). Although we describe a positive role for β-catenin activation during HSV-1 infection here, a recent report described an inhibitory role for the germline transcription factor DUX4 during HSV-1 infection (*Full et al., 2019*). Thus, future studies will need to tease apart the differential contributions and effects of different developmental pathways that are activated during infection.

β-catenin and other developmental pathways are often found to be mutated or dysregulated during tumorigenesis (*Reya and Clevers, 2005*; *Krausova and Korinek, 2014*; *Duchartre et al., 2016*) and are considered to be promising targets for cancer treatment (*Takebe et al., 2015*). In this context, it is interesting to speculate as to the possible impact of β-catenin activation by HSV-1 on its

use as an oncolytic agent (*Sanchala et al., 2017*; *Watanabe and Goshima, 2018*). At present, the first-line treatment for late-stage melanoma is the use of immune checkpoint inhibitors (*Tracey and Vij, 2019*). These inhibitors revolutionized melanoma treatment, but not all patients respond to them. This heterogeneity was shown to be associated with β-catenin activity in the tumor, where high β-catenin levels negatively correlate with treatment success (*Spranger and Gajewski, 2015*; *Spranger et al., 2015*). Given that an HSV-1 -based oncolytic therapy has been FDA approved for late-stage melanoma (*Pol et al., 2015*), it is tempting to speculate that the high level of β-catenin in melanomas that are resistant to checkpoint inhibitors would serve to augment oncolytic HSV-1 replication and anti-tumor effects, although this of course would have to be assessed carefully in separate studies.

# Materials and methods

**Key resources table**

| Reagent type (species) or resource | Designation | Source or reference | Identifiers | Additional information |
|---|---|---|---|---|
| Cell line (*Homo sapiens*) | HDFn, primary human dermal fibrobalsts | Cascade Biologics | cat #C0045C | |
| Cell line (*Homo sapiens*) | A549 | Sigma-Aldrich | cat #86012804-1VL | |
| Cell line (*Homo sapiens*) | Mel624 | | | Mel624 is a patient-derived melanoma cell-line, generated by the lab of Professor Thomas Gajewski at the Univeristy of Chicago |
| Cell line (*Cercopithecus aethiops*) | Vero | Obtained from the lab of Matthew D Weitzman, University of Pennsylvania | | Used to grow wildtype HSV-1 and for plaque assay |
| Cell line (*Homo sapiens*) | U2OS | Obtained from the lab of Matthew D Weitzman, University of Pennsylvania | | Used to grow δICP0 HSV-1 |
| Strain, strain background (HSV-1) | Wild-type strain 17, ICP4-YFP | *Everett et al., 2003* | | Obtained from the lab of Matthew D Weitzman, University of Pennsylvania |
| Strain, strain background (HSV-1) | δICP0 strain 17, ICP4-YFP | *Everett et al., 2003* | | Obtained from the lab of Matthew D Weitzman, University of Pennsylvania |
| Antibody | β-catenin (mouse monoclonal) | R and D systems | MAB13291-SP | Used for IF 1:400 |
| Antibody | IRF3 (rabbit monoclonal) | Cell Signaling Technologies | cat #11904 | Used for IF 1:200 |
| Sequence-based reagent | QPCR primer ICP4 Fwd | IDT | GCGTCGTCGAGGTCGT | |
| Sequence-based reagent | QPCR primer ICP4 Rev | IDT | CGCGGAGACGGAGGAG | |

*Continued on next page*

*Continued*

| Reagent type (species) or resource | Designation | Source or reference | Identifiers | Additional information |
|---|---|---|---|---|
| Sequence-based reagent | QPCR primer ICP8 Fwd | IDT | CGACAGTAACGCCAGAAGCTC | |
| Sequence-based reagent | QPCR primer ICP8 Rev | IDT | GGAGACAAAGCCCAAGACGG | |
| Sequence-based reagent | QPCR primer gB Fwd | IDT | CACCGCTACTCCCAGTTTATGG | |
| Sequence-based reagent | QPCR primer gB Rev | IDT | CCCTTGGCGTTGATCTTGTC | |
| Sequence-based reagent | QPCR primer UL36 Fwd | IDT | CGGGTCAAAAAGGTATGCGGTGT | |
| Sequence-based reagent | QPCR primer UL36 Rev | IDT | TGTCGTACACGCTCCTAACCATTG | |
| Sequence-based reagent | QPCR primer IFIT1 Fwd | IDT | CCT CCT TGG GTT CGT CTA CA | |
| Sequence-based reagent | QPCR primer IFIT1 Rev | IDT | GAA ATG AAA TGT GAA AGT GGC TGA T | |
| Sequence-based reagent | QPCR primer IFIT2 Fwd | IDT | GCTGAATCCTGACAACCAGTACC | |
| Sequence-based reagent | QPCR primer IFIT2 Rev | IDT | CACCTTCCTCTTCACCTTCTTCAC | |
| Sequence-based reagent | QPCR primer CTNNB1 Fwd | IDT | GAGATGGCCCAGAATGCAGTT | |
| Sequence-based reagent | QPCR primer CTNNB1 Rev | IDT | GGTGCATGATTTGCGGGAC | |
| Sequence-based reagent | siRNA against β-catenin | Dharmacon | M-003482-00-0005 | |
| Sequence-based reagent | siRNA non-targeting | Dharmacon | D-001206-13-05 | |
| Commercial assay or kit | RNEasy PLUS minikit | QIAGEN | cat #74134 | |
| Chemical compound, drug | iCRT14 | Sigma-Aldrich | cat ##SML0203 | Stock made in DMSO - 20 mM. Used at 20 micromolar final concentration |
| Software, algorithm | Single cell RNA seq analysis | This paper | https://github.com/ nirdrayman/single-cell-RNAseq-HSV1.git | |

## Cells, viruses and inhibitors

Primary neonatal human dermal fibroblasts (HDFn) were purchased from Cascade Biologics (cat #C0045C), grown and maintained in medium 106 (Cascade Biologics, cat #M106500) supplemented with Low Serum Growth Supplement (Cascade Biologics, cat #S00310). Cells were maintained for upto eight passages, and experiments were performed on cells between passages 4 and 7. A549 cells were purchased from Sigma-Aldrich and maintained in DMEM supplemented with 10% fetal bovine serum. Mel624, a patient-derived melanoma cell-line, was obtained from the lab of Professor Thomas Gajewski at the Univeristy of Chicago and maintained in RPMI supplemented with HEPES, NEAA, Pen/Strep and 10% fetal bovine serum. Vero and U2OS cells (obtained from the laboratory of Matthew D. Weitzman, University of Pennsylvania) were grown in DMEM supplemented with 10% fetal bovine serum and were used for viral propagation and titration. All of the cells that were used were routinely subjected to mycoplasma testing by PCR and found negative.

Wildtype and ΔICP0 HSV-1 (strain 17) viruses expressing ICP4-YFP were generated by Roger Everett (*Everett et al., 2003*) and were a kind gift from Matthew D. Weitzman. Viral stocks were prepared by infecting Vero cells (for wildtype virus) or U2OS cells (for ΔICP0) at an MOI of 0.01. Viral progeny were harvested 2–3 days later using three cycles of freezing and thawing. Viral stocks were

titrated by plaque assays on Vero cells, aliquoted and stored at −80°C. iCRT14, a β-catenin inhibitor, was purchased from Sigma-Aldrich (cat #SML0203) and dissolved in DMSO to make a 20 mM stock solution. iCRT14 stock solution (or DMSO alone as a control) was diluted 1:1000 in growth medium for cell treatment (20 μM final concentration).

## Measuring genomes to plaque-forming unit (PFU) ratio and determining MOI for experiments

We determined the amount of viral genomes in our viral stocks using digital droplet PCR (ddPCR), a method that allows absolute quantification of nucleic acids. 10 μl of viral stock was combined with 90 μl of lysis solution (0.6% SDS, 400 μg/ml Proteinase K) and incubated over night at 37°C. The solution was then boiled (at 95°C) for 10 min and 10-fold serial dilutions were made in $H_2O$. Three primer sets (detecting the viral DNA of the TK, gB or UL36 genes) were used to quantify the amount of viral DNA. PFU were counted by plaque assay on Vero cells. These measurements revealed a genomes: PFU ratio of 36 ± 4 for wildtype HSV-1 and 1,422 ± 34 for ΔICP0. The MOI for experiments was determined empirically, to achieve ~50% ICP4+ cells at 5 hr post infection of HDFn. This corresponded to an MOI of 2 for wild-type virus and an MOI of 0.5 for ΔICP0. Note that, assuming a Poisson distribution, it is unlikely that any of the cells in our experiment did not encounter at least one viral genome (p=$4 \times 10^{-31}$ for wildtype and $1 \times 10^{-304}$ for ΔICP0).

## Time-lapse fluorescent imaging

HDFn cells were seeded on 6-well plates and allowed to attach and grow for one day. On the day of the experiment, cells were counted and infected with HSV-1 at an MOI of 2. Cells were washed once with 106 medium without supplements, and virus was added in the same serum-free media at a final volume of 300 μl per well. Virus was allowed to adsorb to cells for one hour at 37°C with occasional agitation to avoid cell drying. The inoculum was aspirated and 2 ml of full- growth medium was added: this point was considered as 'time zero'. Cells were imaged on a Nikon Ti-Eclipse, which was equipped with a humidity and temperature control chamber. Images were acquired every 15 min for 24 hr from multiple fields of view. Image analysis was performed with ImageJ and MATLAB.

## Single-cell RNA-sequencing

HDFn infected with wildtype HSV-1 at an MOI of 2 or ΔICP0 at an MOI of 0.5 were harvested at 5 hr post-infection and washed three times in PBS containing 0.01% BSA. Cells were counted and processed according to the Drop-seq protocol (*Macosko et al., 2015*) in the Genomics facility core at the University of Chicago. Sequencing was performed on the Illumina NextSeq500 platform. Preliminary data analysis (quality control, trimming of adaptor sequences, UMI and cell barcode extraction) was performed on a Linux platform using the Drop-seq Tools (Version 1.13) and the Drop-seq Alignment Cookbook (Version 1.2), which are available at https://github.com/broadinstitute/Drop-seq/releases. Alignment of reads was performed using the STAR aligner (Version 2.5.4b) (*Dobin et al., 2013*) to a concatenated version of the human GRCh38 primary assembly (Gencode release 27) and HSV-1 genomes (Genbank accession: JN555585). The HSV-1 genome annotation file was kindly provided by Moriah Szpara (Pennsylvannia State University). Following the generation of the DGE (digital gene expression) file, further analyses were performed in MATLAB, these included quality control, cell clustering, correlation and differential gene expression analyses and data visualization. All the of the scripts used for data analysis have been deposited in Github (*Drayman, 2019*; copy archived at https://github.com/elifesciences-publications/single-cell-RNAseq-HSV1). Key points in the analysis are expanded on below.

### Cell filtering

Following the construction of the initial DGEs (digital expression matrices), we filtered out cells with low (below 2000) and high (above 10,000) Unique Molecular Identifier (UMI) counts. We then assessed the fraction of mitochondrial genes in individual cells and filtered out cells with high mitochondrial fraction (above 0.2 for mock- and wildtype-infected cells, above 0.4 for ΔICP0-infected cells). The distributions of the number of UMI, total genes and mitochondrial genes are presented in *Figure 1—figure supplement 1*. This resulted in three DGEs, one for each condition (mock-infected, wildtype-infected and ΔICP0-infected) containing 4500, 807 and 1613 cells, respectively.

### Normalization and UMI regression

The DGEs were log-normalized by dividing gene expression by the total number of UMI for each cell, multiplying by 10,000, adding one and taking the log of the value. We then regressed out the effect of the number of UMI on gene expression, by constructing a linear model for each gene's expression as a function of total number of UMI. The expected value from the model was subtracted from the gene's expression and the residual kept. We then performed Z-scoring, by subtracting the gene's mean expression and dividing by the gene's standard deviation.

### Cell-cycle regression

A list of 14 $G_2/M$ marker genes (HMGB2, CDK1, NUSAP1, UBE2C, BIRC5, TPX2, TOP2A, NDC80, CKS2, NUF2, CKS1B, MKI67, TMPO, and CENPF) was used to construct a cell-cycle score (for every gene from the list expressed by the cell, +1 was added to the score). Regression of the cell-cycle score was done as described above for UMI regression (keeping the residual after linear model fitting).

### Clustering and differential gene expression analysis

Cell were clustered using k-means clustering. For clustering wild-type-infected cells, we used all the host and viral genes. For ΔICP0-infected cells, we used the viral genes as well as the top host genes that correlated with viral gene expression (top 100 correlated and top 100 anti-correlated). Genes that are differentially expressed between clusters were then identified by a two-sided Wilcoxon rank sum test, followed by Benjamini and Hochberg false-detection rate (FDR) correction. A gene was considered differentially expressed if the FDR-corrected p-value was below 0.05.

## RNA-sequencing of sorted cells

HDFn cells were mock infected or HSV-1 infected as described above, trypsinized, washed and re-suspended in full growth media. Cells were filtered through a 100 µm mesh into FACS sorting tubes and kept on ice. HSV-1 infected cells were sorted into two populations based on their ICP4-YFP expression. 0.5 million cells were collected from each population. Mock-infected cells were similarly sorted. ICP4-negative cells had the same level of YFP fluorescence at mock-infected cells. For ICP4-positive cells, we collected cells that were in the top 30% of YFP expression. The two populations were clearly separated from each other. Sorting was performed on an AriaFusion FACS machine (BD) at the University of Chicago flow-cytometry core facility. Total RNA was extracted from cells using the RNeasy Plus Mini Kit (QIAGEN) and submitted to The University of Chicago Genomics core for library preparation and sequencing on a HiSeq4000 platform (Illumina). Reads were mapped to a concatenated version of the human and HSV-1 genomes with STAR aligner (see single-cell RNA-sequencing above for details). Reads were counted using the featureCounts command, which is a part of the Subread package (*Liao et al., 2013*). Further analyses were performed in MATLAB and these included differential gene expression analyses and data visualization.

## Data availability

All sequencing data have been deposited in the Gene Expression Omnibus (GEO) under accession number GSE126042. All of the scripts used for data analysis and visualization are available through GitHub at: https://github.com/nirdrayman/single-cell-RNAseq-HSV1.git.

## Immunofluorescence staining

HDFn were seeded in 24-well plates and allowed to attach and grow for one day. Cells were infected as described above and fixed using a 4% paraformaldehyde solution at 5 hr post-infection. Cells were fixed for 15 min at room temperature and washed, blocked and permeabilized with a 10% BSA, 0.5% Triton-X solution in PBS for one hour. Cells were then incubated with primary antibodies in a staining solution (2% BSA, 0.1% Triton-X in PBS) overnight at 4°C. Cells were washed three times with PBS, incubated with secondary antibodies in staining solution for 1 hr at room temperature, washed three times with PBS and covered with 1 ml PBS containing a 1:10,000 dilution of Hoechst 33342 (Invitrogen, cat #H3570). Cells were imaged on a Nikon Ti-Eclipse inverted epi-fluorescent microscope. Primary antibodies were mouse monoclonal anti-β-catenin (R and D systems, cat #MAB13291, used at 1:200 dilution) and rabbit monoclonal anti-IRF3 (Cell Signaling Technologies,

Cat #11904S, used at 1:400 dilution). Secondary antibodies were AlexaFluor 555 conjugated anti-mouse and anti-rabbit F(ab')two fragments (Cell Signaling Technologies, cat #4409S, #4413S, used at 1:1000 dilution).

## siRNA nucleofection

$5 \times 10^5$ HDFn cells were washed once in PBS and nucleofected with 1 µM siRNA against β-catenin (Dharmacon, siGENOME Human CTNNB1, cat #M-003482-00-0005) or with a scrambled siRNA control (Dharmacon siGENOME Non-Targeting siRNA Pool #1, cat #D-001206-13-05) using the Human Dermal Fibroblast Nucleofector Kit (Lonza, cat #VPD-1001). β-catenin expression was assayed 3 days later by Q-PCR.

## Acknowledgements

We wish to thank Matthew D Weitzman for sharing with us the ICP4-YFP expressing HSV-1 and Moriah Szpara for the genome annotation of the HSV-1 strain 17. Sequencing and library preparations were performed at The University of Chicago Genomics core facility and cell sorting at the Flow Cytometry core. We also with to thank Oren Kobiler for support and advice throughout the project. ND wishes to thank EMBO and HFSPO for their support through post-doctoral fellowships at different stages of the project.

## Additional information

### Funding

| Funder | Grant reference number | Author |
| --- | --- | --- |
| Human Frontier Science Program | post-doctoral fellowship | Nir Drayman |

The funders had no role in study design, data collection and interpretation, or the decision to submit the work for publication.

### Author contributions

Nir Drayman, Conceptualization, Formal analysis, Investigation, Visualization, Methodology, Writing—original draft, Writing—review and editing; Parthiv Patel, Investigation; Luke Vistain, Investigation, Writing—original draft; Savaş Tay, Conceptualization, Supervision, Writing—original draft, Writing—review and editing

### Author ORCIDs

Nir Drayman https://orcid.org/0000-0003-4460-9558

### Decision letter and Author response

Decision letter https://doi.org/10.7554/eLife.46339.026
Author response https://doi.org/10.7554/eLife.46339.027

## Additional files

### Supplementary files

• Supplementary file 1. Differential gene expression identified by sRNAseq. (A) Genes that are upregulated in highly infected cells (wildtype infection). (B) GO annotations associated with genes from tab (A). (C) Transcription factors enriched in the promoters of genes from tab (A). (D) Genes that are upregulated in highly infected cells (ΔICP0 infection). (E) GO annotations associated with genes from tab (D). (F) Transcription factors enriched in the promoters of genes from tab (D).
DOI: https://doi.org/10.7554/eLife.46339.019

• Supplementary file 2. Analysis of genes that are upregulated in ICP4-negative sorted cells. (A) Genes that are upregulated in ICP4-negative cells (afer wildtype infection). (B) GO annotations

associated with genes from tab (A). (C) Genes that are upregulated in ICP4-negative cells (after ΔICP0 infection). (D) GO annotations associated with genes from tab (C). (E) Transcription factors enriched in the promoters of genes from tab (C).

DOI: https://doi.org/10.7554/eLife.46339.020

• Supplementary file 3. Analysis of genes upregulated in ICP4-positive sorted cells. (A) Genes that are upregulated in ICP4-positive cells (after wildtype infection). (B) GO annotations associated with genes from tab (A). (C) Transcription factors that are enriched in the promoters of genes from tab (A). (D) Genes that are upregulated in ICP4-positive cells (after ΔICP0 infection). (E) GO annotations associated with genes from tab (D). (F) Transcription factors that are enriched in the promoters of genes from tab (D).

DOI: https://doi.org/10.7554/eLife.46339.021

• Transparent reporting form

DOI: https://doi.org/10.7554/eLife.46339.022

## Data availability

All sequencing data has been deposited in the Gene Expression Omnibus (GEO) under accession number GSE126042. All the scripts used for data analysis and visualization are available through GitHub at: https://github.com/nirdrayman/single-cell-RNAseq-HSV1.git (copy archived at https://github.com/elifesciences-publications/single-cell-RNAseq-HSV1).

The following dataset was generated:

| Author(s) | Year | Dataset title | Dataset URL | Database and Identifier |
|---|---|---|---|---|
| Drayman N, Patel P, Vistain L, Tay S | 2019 | Single cell analysis of HSV-1 infection reveals anti-viral and developmental programs are activated in distinct sub-populations with opposite outcomes | https://www.ncbi.nlm.nih.gov/geo/query/acc.cgi?acc=GSE126042 | NCBI Gene Expression Omnibus, GSE126042 |

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
