## [Decision Letter]

Thank you for submitting your article "HSV-1 single cell analysis reveals anti-viral and developmental programs activation in distinct sub-populations" for consideration by *eLife*. Your article has been reviewed by three peer reviewers, and the evaluation has been overseen by a Reviewing Editor and Naama Barkai as the Senior Editor. The following individual involved in review of your submission have agreed to reveal their identity: Jesse D Bloom (Reviewer #1).

The reviewers have discussed the reviews with one another and the Reviewing Editor has drafted this decision to help you prepare a revised submission.

Summary:

Single cell analyses provide the unprecedented opportunity to study virus-host interactions while exploring and capturing the heterogeneity of cell responses. Although these studies have focused mainly on RNA viruses, the present paper focuses on the Herpex simplex virus 1 (HSV-1) DNA virus. They find that there is substantial heterogeneity among cells, the expression of interferon-stimulated genes is rare, and infection induces a developmental program in the cells. The study is interesting and well done, and the paper is well written.

Essential revisions:

1) All reviewers had questions about the population of uninfected cells and the discussion thereof.

- Subsection “Viral infection dynamics varies among individual cells”, first paragraph: how was the MOI titered? Only 50% of cells become ICP4-positive at a MOI of 2. This is extremely far off Poisson statistics, which would suggest that 1 – exp(-2) = 84% should be ICP4-positive. The aforementioned paragraph just shows that infection isn't Poisson suggesting some cells might be refractory to infection. This should be more explicitly discussed at this point.

- Subsection “Viral infection dynamics varies among individual cells”, third paragraph and Figure 1C: Clarity of the wording is important: cells can be exposed but not infected (the virus entered); cells can be abortively infected, i.e. the virus entered but cannot successfully replicate and cells can undergo productive infection. Thus, being ICP4- during imaging analysis does not allow discriminating between abortive and non-infected yet. In this context, I would rather write "of 1,814 cells exposed to HSV-1, […]".

- In Figure 1, only ICP4^+^ cells are shown and it is not clearly stated what criteria were used to distinguish ICP4^+^ and ICP4^-^ cells, nor is the distribution for ICP4^-^ cells ever shown. Please provide the intensity of the 818 ICP4^-^ cells as well as their number. Then, after the scRNA-Seq, please include your interpretation that these data indicate that the majority of cells are expressing some% of viral transcripts and thus are bona fide abortive infections rather than non-infected cells (as only apparently a minor fraction of cells display 0% viral transcripts),

2) Provision of data and methods:

-The authors should add a figure supplement that shows some basic statistics on read depth per cell (and its distribution) for each sample. This is critical for evaluating single-cell RNA-seq, and doesn't appear possible to find now short of going to the raw data.

- The Materials and methods say that the scripts for data analysis are "available upon request." This is not adequate for a paper that relies so heavily on computational analyses. The authors need to provide the computer code, which is currently not contained as part of the paper. The scripts should be made available (*eLife* allows GitHub repos linked to papers), and the paper should not be accepted until these scripts along with some sort of reasonable README describing their use are available for examination. In addition, some further details (such as how the differential gene expression was done) should be added to the Materials and methods.

3) The very interesting idea of the role of the cell cycle in infection and signaling should be further developed rather than just mentioned as a possibility, given the richness of the data at hand. The scRNA-seq should allow the testing of correlation between viral phenotype and cell cycle. In the subsection “The cell-cycle affects HSV-1 gene expression” and Figure 2—figure supplement 1, please provide more details and results regarding the cell cycle score and how the outregression was performed (gene list used for the G2/M score and outregression).

Major comments:

1) Please include references to relevant papers such as Zhu and Jones, 2018, which shows that HSV1 viral production is inhibited upon treatment with iCRT14. It would be good also to cite other scRNA-seq papers on the sparsity of IFN induction in virus-infected cells, such as DOI 10.1128/JVI.01778-18 and DOI 10.1101/437277.

2) The expression of ISGs like IFIT1 and Mx2 which are produced in response to IFN-receptor signaling are most frequently analyzed. IRF3 activation is also examined and it would be helpful if the authors would discuss whether that was indicative of the cell-autonomous response, as one might conclude. This would help shed light on whether cells are detecting infection or just responding to signaling.

3) The argument that ISG expression is higher among low-virus expressing cells infected by deltaICP0 is not entirely convincing. It is true that in Figure 3D, all of the cells expressing ISGs are in clusters 1-3 (except two IFIT1-expressing cells in cluster 4 which are not accurately described in the text which says that no cluster 4 cells express ISGs). However, cluster 4 has many fewer cells, so maybe there just aren’t ISG expressing cells in that cluster by chance. Some statistics should be applied here. Also, Figure 2A, B, C appears to show a IFIT2 expressing cells with high viral expression. Please show additional data to strengthen the conclusions if available or obtainable. It is not clear whether the cell clustering analysis was performed taking into account viral transcripts or not. Could you please compare cell clustering with mock cells performed in parallel?

---

## [Author Response]

Essential revisions:1) All reviewers had questions about the population of uninfected cells and the discussion thereof.- Subsection “Viral infection dynamics varies among individual cells”, first paragraph: how was the MOI titered? Only 50% of cells become ICP4-positive at a MOI of 2. This is extremely far off Poisson statistics, which would suggest that 1 – exp(-2) = 84% should be ICP4-positive. The aforementioned paragraph just shows that infection isn't Poisson suggesting some cells might be refractory to infection. This should be more explicitly discussed at this point.

The virus was titrated on Vero cells, which are more permissive for HSV-1 infection than HDFn cells. An MOI of 2 (based on Vero titration) was chosen as it empirically resulted in ~50% of cells becoming ICP4^+^. We have amended the manuscript to reflect this. The relevant part now reads: “HDFn were infected at an MOI of 2 (calculated based on virus titration on Vero cells, which are ~2-fold more susceptible to HSV-1 infection than HDFn). […] Note that we determined the genome:PFU ratio for our viral stock and found it to be 36 ± 4, suggesting that all the cells in the culture have likely encountered numerous virus particles.”

We have also added a section to the Materials and methods to make it clearer:

**“**Measuring genomes to PFU ratio and determining MOI for experiments: We determined the amount of viral genomes in our viral stocks using digital droplet PCR (ddPCR), a method that allows absolute quantification of nucleic acids. […] Note that, assuming a Poisson distribution, it is unlikely that any cells in our experiment did not encounter at least one viral genome (p=4X10^-31^ for wt and 1X10^-304^ for ΔICP0).”

- Subsection “Viral infection dynamics varies among individual cells”, third paragraph and Figure 1C: Clarity of the wording is important: cells can be exposed but not infected (the virus entered); cells can be abortively infected, i.e. the virus entered but cannot successfully replicate and cells can undergo productive infection. Thus, being ICP4^-^ during imaging analysis does not allow discriminating between abortive and non-infected yet. In this context, I would rather write "of 1,814 cells exposed to HSV-1, [...]"

We have amended this part which now reads: “HDFn were infected as above, fixed and stained with DAPI at 5 hours post-infection (to allow automated cell segmentation and quantification). This allowed the distinction of two cellular populations: cells which successfully initiated viral gene expression (ICP4^+^) and cells which did not (ICP4^-^). Of 1,814 cells exposed to HSV-1, 996 cells (55%) were ICP4^+^ and 818 (45%) were ICP4^-^. Cells were classified as ICP4 negative or positive based on a threshold calculated from mock-infected cells (mean+3 standard deviations).”

We have also amended Figure 1C as described below.

*- In Figure 1, only* ICP4^+^
*cells are shown and it is not clearly stated what criteria were used to distinguish* ICP4^+^
*and* ICP4^-^
*cells, nor is the distribution for* ICP4^-^
*cells ever shown. Please provide the intensity of the 818* ICP4^-^
*cells as well as their number.*

We have amended Figure 1C to show the ICP4 levels of mock, ICP4^-^ and ICP4^+^ cells. We have also changed the display from a violin plot to also show the values measured in individual cells (as circles). Since mock and ICP4^-^ cells have dramatically lower ICP4 levels than ICP4^+^ cells (which is basically fluorescence background), we have included a zoomed-in view of these populations as an inset.

We have added a description of the criteria for classifying cells as ICP4 negative or positive: “Cells were classified as ICP4 negative or positive based on a threshold calculated from mock-infected cells (mean+3 standard deviations)”.

Then, after the scRNA-Seq, please include your interpretation that these data indicate that the majority of cells are expressing some% of viral transcripts and thus are bona fide abortive infections rather than non-infected cells (as only apparently a minor fraction of cells display 0% viral transcripts),

We have added this to the text: “The vast majority of cells exposed to HSV-1, either wt or ΔICP0, had some level of viral gene expression, suggesting that the fraction of lowly-expressing cells (and the above noted ICP4^-^ population) are indeed abortively-infected cells, rather than cells that did not encounter a virus.”

2) Provision of data and methods:-The authors should add a figure supplement that shows some basic statistics on read depth per cell (and its distribution) for each sample. This is critical for evaluating single-cell RNA-seq, and doesn't appear possible to find now short of going to the raw data.

We have added these data including filtering criteria, UMI per cell, gene per cell and mitochondrial fraction per cell as a new Figure 1—figure supplement 1. We have also added this to the text: “Technical data on sequencing depth and filtering criteria are presented is Figure 1—figure supplement 1.”

- The Materials and methods say that the scripts for data analysis are "available upon request." This is not adequate for a paper that relies so heavily on computational analyses. The authors need to provide the computer code, which is currently not contained as part of the paper. The scripts should be made available (eLife allows GitHub repos linked to papers), and the paper should not be accepted until these scripts along with some sort of reasonable README describing their use are available for examination.

We have deposited the code as well as a README file in GitHub and have linked to the repository in the Materials and methods: “All the scripts used for data analysis and visualization are available through GitHub at: https://github.com/nirdrayman/single-cell-RNAseq-HSV1.git.”. We tried to make the code as readable as possible, and have included in the GitHub repository a matlab file that holds all the initial sequencing data required for analysis. The script that accompanies the data can be run directly on these files to reproduce our results.

We have also changed the status of our GEO submission (containing all RNA-sequencing data) from private to public and it is now accessible to the general public.

In addition, some further details (such as how the differential gene expression was done) should be added to the Materials and methods.

We have added the description of criteria for cell filtering, data normalization, UMI and cell-cycle regression, clustering and differential gene expression identification to the Materials and methods section:

“Cell filtering: Following the construction of the initial DGEs (digital expression matrices), we filtered out cells with low (below 2,000) and high (above 10,000) UMI (Unique Molecular Identifier) counts. […] A gene was considered differentially expressed if the FDR-corrected p-value was below 0.05.”

3) The very interesting idea of the role of the cell cycle in infection and signaling should be further developed rather than just mentioned as a possibility, given the richness of the data at hand. The scRNA-seq should allow the testing of correlation between viral phenotype and cell cycle. In the subsection “The cell-cycle affects HSV-1 gene expression” and Figure 2—figure supplement 1, please provide more details and results regarding the cell cycle score and how the outregression was performed (gene list used for the G2/M score and outregression).

We now included a more detailed description of the cell-cycle effect: “We have previously shown that the cell-cycle position at the time of infection is a cellular determinant of HSV-1 successful infection in the H1299 cell-line (Drayman et al., 2017). […] We found that viral gene expression is negatively correlated with the cell-cycle score, with cells in the later parts of the cell-cycle expressing ~10-fold less viral genes than those in the early part of the cycle, in agreement with our previous finding in the H1299 cell-line.”

We have also provide a description for the cell-cycle score calculation and its out-regression in the Materials and methods section: “Cell-cycle regression: A list of 14 G2/M marker genes (HMGB2, CDK1, NUSAP1, UBE2C, BIRC5, TPX2, TOP2A, NDC80, CKS2, NUF2, CKS1B, MKI67, TMPO, CENPF) was used to construct a cell-cycle score (for every gene from the list expressed by the cell, +1 to the score). Regression of the cell-cycle score was done as described above for UMI regression (keeping the residual after linear model fitting).”

Major comments:1) Please include references to relevant papers such as Zhu and Jones, 2018, which shows that HSV1 viral production is inhibited upon treatment with iCRT14. It would be good also to cite other scRNA-seq papers on the sparsity of IFN induction in virus-infected cells, such as DOI 10.1128/JVI.01778-18 and DOI 10.1101/437277.

We have added the suggested references.

“In accordance with its impact on late viral gene expression, β-catenin inhibition significantly reduced viral progeny formation (Figure 7F). This is in agreement with a recent report by Zhu and Jones showing iCRT14 reduced plaque formation in two other cell types (HLF and Vero) (Zhu and Jones, 2018).”

“This rare activation of anti-viral signaling seems to be widespread, as single-cell investigations into RNA viruses such as West-Nile virus (O’Neal et al., 2019) and Influenza (Russell et al., 2018a, 2018b) have reported similar findings.”

2) The expression of ISGs like IFIT1 and Mx2 which are produced in response to IFN-receptor signaling are most frequently analyzed. IRF3 activation is also examined and it would be helpful if the authors would discuss whether that was indicative of the cell-autonomous response, as one might conclude. This would help shed light on whether cells are detecting infection or just responding to signaling.

We have clarified this point: “As previous population-level studies reported the activation of anti-viral genes during wild-type HSV-1 infection, we hypothesized that highly infected cells (Figure 2A, B, cluster 1) should be enriched for anti-viral genes. […] We thus conclude that wt HSV-1 infection efficiently blocks the induction of the anti-viral response and that the activation of anti-viral genes in population averaged measurements is the result of anti-viral signaling elicited in a rare population of abortively infected cells.”

3) The argument that ISG expression is higher among low-virus expressing cells infected by deltaICP0 is not entirely convincing. It is true that in Figure 3D, all of the cells expressing ISGs are in clusters 1-3 (except two IFIT1-expressing cells in cluster 4 which are not accurately described in the text which says that no cluster 4 cells express ISGs). However, cluster 4 has many fewer cells, so maybe there just aren’t ISG expressing cells in that cluster by chance. Some statistics should be applied here.

We have now more accurately described the data: “These cells had low viral gene expression levels and mostly belonged to clusters 1-3 (Figure 3C, D). Anti-viral signaling was not seen in highly-infected cells of cluster 4, with the exception of two cells expressing *IFIT1* (Figure 3C, D).”

We have now included p-values (from a Chi-square test) in the figure, showing that anti-viral gene expression by cells in cluster 4 is indeed significantly different than clusters 1-3.

Furthermore, to overcome the problem of different group sizes, we have performed a complementary analysis (now Figure 3—figure supplement 1), where we have compared two equal sized groups – taking the top and bottom 25% of cells, according to their HSV-1 gene expression. We have analyzed 23 ISGs, 21 of which show a significant decrease in expression in the cells with high HSV-1 gene expression, supporting our initial conclusion. This is described as follows: “As we have described above for the wild-type infected cells, we also compared a larger panel of ISGs in high vs low infected cells (Figure 3—figure supplement 1), with similar results.”

Also, Figure 2A, B, C appears to show a IFIT2 expressing cells with high viral expression. Please show additional data to strengthen the conclusions if available or obtainable.

This data is now presented in the new Figure 2—figure supplement 2 and is described as follows: “When comparing a larger panel of ISGs (Interferon Stimulated Genes) in high vs low infected cells (Figure 2—figure supplement 2), most ISGs are in fact more highly expressed in cells with low HSV-1 gene expression.”

It is not clear whether the cell clustering analysis was performed taking into account viral transcripts or not. Could you please compare cell clustering with mock cells performed in parallel?

We used both viral and host genes for clustering. We have now included a description of clustering in the Materials and methods: “Clustering and differential gene expression analysis: Cell were clustered using k-means clustering. […] A gene was considered differentially expressed if the FDR-corrected p-value was below 0.05.”

This is also stated in the figure legends.

In response to the reviewers’ request, we have performed a joint analysis of mock and HSV-1 infected cells, taking into account both host and viral genes. The results show that highly-infected cells cluster separately from the rest of the cells, with low-infected and mock-infected cells clustering together. This is reported as follows: “The viral gene expression distribution was highly skewed, with most cells expressing low levels of viral transcripts and some cells expressing much higher levels (Figure 1E). A joint analysis of mock and wt-infected cells shows that highly-infected cells cluster separately from the mock-infected and low-infected cells, which are intermingled (Figure 1—figure supplement 2).”